# Optimizing skin cancer screening with convolutional neural networks in smart healthcare systems

**Ali Raza**[1], **Akhtar Ali**[1]*, **Sami Ullah**[2], **Yasir Nadeem Anjum**[3], **Basit Rehman**[1]

**1** Department of Mathematics, Government College University, Faisalabad, Pakistan, **2** Department of Computer Science, Government College University, Faisalabad, Pakistan, **3** Department of Applied Sciences, National Textile University, Faisalabad, Pakistan

* utm.akhtar@gmail.com

**Data availability statement:** The raw image dataset used in this study is publicly available

## Abstract

Skin cancer is among the most prevalent types of malignancy all over the global and is strongly associated with the patient's prognosis and the accuracy of the initial diagnosis. Clinical examination of skin lesions is a key aspect that is important in the assessment of skin disease but comes with some drawbacks mainly with interpretational aspects, time-consuming and healthare expenditure. Skin cancer if detected early and treated in time can be controlled and its deadly impacts arrested completely. Algorithms applied in convolutional neural network (*CNN*) could lead to an enhanced speed of identifying and distinguishing a disease, which in turn leads to early detection and treatment. So as to eliminate these challenges, optimized *CNN* prediction models for cancer skin classification is studied in this researche. The objectives of this study were to develop reliable optimized *CNN* prediction models for skin cancer classification, to handle the severe class imbalance problem where skin cancer class was found to be much smaller than the healthy class. To evaluate model interpretability and to develop an end-to-end smart healthcare system using explainable AI (*XAI*) such as *Grad-CAM* and *Grad-CAM++*. In this researche new activation function namely *NGNDG-AF* was offered specifically to enhance the capabilities of network fitting and generalization ability, convergence rate and reduction in mathematical computational cost. A research used an optimized *CNN* and *ResNet152V2* with the *HAM10000* dataset to differentiate between the seven forms of skin cancer. Model training involved the use of two optimization functions (*RMSprop* and *Adam*) and *NGNDG-AF* activation functions. Cross validation technique the holdout validation is used to estimate of the model's generalization performance for unseed data. Optimized CNN is performing well as compare to ResNet152V2 for unseen data. The efficacy of the optimized *CNN* method with *NGNDG-AF* was examined by a comparative study wirh popular *CNN* with various activation functions shows that better performance of *NGNDG-AF*, achieving the classification accuracy rates that are as high as 99% in training and 98% in the validation. The recommended system also involves the integration of the smart healthcare application as a central component to give the doctors

on: https://www.kaggle.com/datasets/kmader/skin-cancer-mnist-ham10000 (DOI: 10.7910/DVN/DBW86T). All relevant data supporting the findings of this study are included within the manuscript.

**Funding:** The author(s) received no specific funding for this work.

**Competing interests:** The authors have declared that no competing interests exist.

as well as the healthcare providers diagnosing and tools that would assist in the early detection of skin cancer hence leading to better outcomes of the treatment.

## 1 Introduction

The development of skin cancer is possible when cells of the skin are destroyed like excessive exposure to sunlight which produces ultra violet (UV) radiation [1]. Unlike many other malignancies, skin cancer cases are not compulsory for inclusion in global cancer statistics [2]. However, according to WHO, one out of three cancers that are diagnosed is a skin cancer case [3]. The exploration of computer vision and deep learning as a way to eliminate skin cancer has various significant advantages. These technologies enable early diagnosis by automatically analyzing large amounts of medical data including dermoscopic images. This could result in quicker interventions, better patient outcomes, cost savings in healthcare provision and improved access to expert diagnostics particularly in areas with limited healthcare resources. Earlier studies have ignored what can be achieved with explainable AI technology choosing only computer vision and deep learning models. Compared to these early works which have concentrated on only computer vision and deep learning solutions for classifying cancer images, our research includes explainable AI technology together with advanced computer vision and deep learning approaches. Thorough fusion allows us to classify cancer images accurately as well as offering comprehensive insights into the model's decision making process. This integration also improves the interpretability, transparency, and reliability of our approach so that we can achieve more accurate and clinically useful results in medical image analysis. Machine learning based automated detection technique is one way through which this problem can effectively be solved. The increasing popularity of deep learning in medical image processing is attributed to its ability to autonomously learn complex features from raw data. Many studies have heavily relied on these approaches in their research such as those focusing on cardiovascular diseases classification, cancer prediction [4] pneumonia diagnosis among others [5]. The HAM10000 dataset was used in the study to assess two CNN models: ResNet152V2 [6] and the optimized CNN. Notably, the optimized CNN surpassed the other models in the experiment, achieving a validation accuracy rate of 98.12%. A multitude of Explainable Artificial Intelligence systems have been developed for image categorization. Yet, the identification and classification of skin cancer have not received as much focus. In this paper, Grad-cam [7] Grad-Cam++ [8] were employed to accurately classify the images. In this paper, an optimized Convolutional Neural Network (CNN) model is employed on the HAM10000 [9] dataset for the identification of seven types of skin lesions. To improve the explanatory power and accuracy of the model Grad-CAM, Grad-CAM++ were used in this study. The training was done at great length to handle imbalanced datasets and its impact on model accuracy has been shown.

**The main contributions of this paper are as follows:**

- Introduction of an optimized CNN model and application of Grad-CAM and Grad-CAM++ techniques equipped with the NGNDG-AF activation function, further enhancing the model's capabilities.
- At several network layers, the appropriateness of the proposed NGNDG-AF is evaluated.
- Addressing imbalanced datasets through extensive training.
- Utilization of an optimized CNN model on the HAM10000 dataset for skin lesion identification.

- Integration of Grad-CAM, Grad-CAM++ techniques to enhance model explanation and accuracy.

These contributions collectively lead to a more reliable and effective model for diagnosing skin cancer, which is crucial for improving healthcare outcomes in dermatology.

## 2 Literature review

Saturating functions such as sigmoid and hyperbolic functions were originally used to activate neurons in Artificial Neural Networks (ANNs) [10–13] However, it becomes very difficult to transmit the error rate back across the network if any unit starts within the saturating region of the activation function. Overfitting and vanishing gradient issues led to the development of ReLU [14]. The positive part of the input passes through unaltered due to ReLU's linear behavior and non-saturating characteristics, which inhibit the negative portion from impacting the output. This highlights the critical role that activation functions play in achieving nonlinear modeling by encouraging linear gradient flow during backpropagation in deep network models. In the 2012 ImageNet ILSVRC competition, ReLU debuted as the activation function [15]. ReLU is a particularly noteworthy and effective advance in deep convolutional neural networks among activation functions. To solve the problem of neuron loss during training, several adjustments were implemented as a result. The solution to this problem was introduced in 2013 with the introduction of Leaky ReLU (LReLU), which treats the positive half of the input similarly to ReLU but assigns a non-zero slope value to the negative component [16]. To enhance the accuracy of the negative signal components, Parametric ReLU (PReLU) in [17,18] introduced the ability to train the slope of the coefficient instead of fixing it at a fixed value. This invention efficiently addresses the issue of neuron death while competing with linear mapping terms for non-linear transformation functions. Negative slope coefficients are trained using model weights throughout the training process. Effectively, the activation pattern of the negative portion of the model is shaped as each layer learns different values for the parameters. The Exponential Linear Unit [19] models uses an exponential function to simulate the negative portion of the input signal. The nonlinear mapping term is forced to compete with the original input for nonlinear transformation capabilities, hence solving the output variance and bias issues. Major drawbacks of ELU is exponential base and neede more mathematically computation during a training where exponential gradient and function utilized to updating parameters. The scaled version of ELU, called SELU [20], forces neuron activations across the network toward unit variance and zero mean. Single layer feed forward neural network's results showed that MEA's value in small range and outperform ReLU and ELU. Average MEA's value 0.76, 0.82 and 0.83 for SELU, ReLU and ELU respectively, listless of optimization algorithms and hyperparameters. Motivated by the intuition that sigmoidal activation functions are insufficient and inefficient for handling real-world recognition and classification tasks, In their work, Aboubakar Nasser, Samatin Njikam and Huan Zhao [21] calculated effectivess of a novel activation function termed the "rectified hyperbolic secant activation function," which shares similarities with the Exponential Linear Unit (ELU). This activation function presumably offers unique characteristics that could potentially enhance the performance of neural networks with data sets MNIST, CIFAR-10, CIFAR-100. Moreover, in another study stated as [22], the authors studied a AbsLU activation function. AbsLU activation function implemented on three different ANN models for data sets CIFAR-100 , CIFAR-10 and fashion MNIST with highest average accuracy 54.47%, 72.51% and 93.98% respectively as compare to other standard activation function. This modification suggested to

make the ReLU activation function to be Leaky ReLU, which means it will return a small negative value and a positive value – may be it will help for certain in terms of model robustness against adversarial examples, or for features learning. Later in 2016, Dan Hendrycks, and Kevin Gimpel presented the activation function that was at first called the Gaussian Error Linear Unit [23]. To express it, this activation function entails the multiplication of each input with the cumulative density of the normal distribution. On the other hand, the Swish activation function, which was offered in 2017 originally by Ramachandran et al. [24] from Google Brain, multiplies the input by the sigmoid function. Swish distinctively is not a strictly monotonic activation function as it first rises and then it falls slightly before rising again in the negative domain. All the same, they notice that this general feature does attract consideration due to the power of Swish. For this reason, to determine the exact activation contour for each layer of DNNs. Trottier et al. [25] in 2017, a new method called Parametric ELU or PELU was developed because the latter requires the learning of the ELU's parameterization. In training, there are several elements and these are normally trained in parallel and these include the biases as well as the weights of PELU . However, in year of 2018, Ömer Faruk Ertuğrul analyzed a refined idea of the utilization of linear regression model in training an ideal activation function [26]. The very next year, that is in the year 2019, mishra et al. studied another activation function called Mish activation function [27]. They trained a new activation function that was superior to ReLU and Swish activation functions when the latter was tested on CIFAR-100 suing the Squeeze Excite Net-18. Thus, the activation of Mish shows relevance in improving the size of neural networks, specifically in issues like the image identification. In a work of 2021, Sayan Nag et al. [28] formulated the activation function known as Serf which is non-monotonically activated and, therefore, has the ability of self-regularisation. Unlike other most well-known linear activation functions include ReLU, sigmoid or among others, the present work reveals that the Serf function has non-monotonous output; this means that Serf function's output response does not mimic the inputs in a percentage form or in a linear manner. Also in this regard, the self-regularization characteristic that fights the inherent tendency of Neural Networks to overfit help in the regularization of the model from the training phase. Wang et al. [29] Only employed Smish activation, in which Smish was used as their nonlinear activation function. This new approach is distinct from the types of activation functions that are usually applied because it helps to control the specific nonlinearity that was introduced in the course of the model's development. It is either bounded from below, or non decreasing and continuous. It can be noted in general that the level of LBA intricacy is high in smish as compared to its competitors. In addition, in the year 2022, Shui-Long Shen and his team [30] studied another activation function it is the tanhLU. This is another creative function where a linear unit is incorporated with hyperbolic tangent function, which in my option is magnificent. The biggest comprehensible difference that has been carried out is the comparison which has been made across five forms of architectures of neural networks and seven forms of benchmark datasets from the different domains of application where it has been realized that tanhLU has better performance as compared to other activation functions. In the year 2023, a completely new type of activation function that is non-monotonic was recommended in the field of computer vision by Iván Vallés-Pérez et al. [31] the activation functions play a very important part in the neural networks, as they act as a way of introducing the non-linearity in the system which helps to capture the complex patterns within the given data sets. It is true that generally there is a slight increase in terms of computational complexity but it exhibited additional generality over some other forms of nonlinearities. Previous activation functions primarily dealt with the negative part of the signal only. That being said, we introduce a novel activation function based on algebraic theory and with low computational complexity. This offered function not only helps us deal with the negative activation aspect but also allows us

to well portray the nonlinearity of the connections in the neuronal maps of the network in the subsequent function. Also known as artificial neural network or multi-layer neural network, deep learning helps the network learn from large amount of data. This paper presented a discussion on few application of deep learning in bioinformatics, the field of study that aims at analyzing and interpreting biological data. Multi-view deep learning model iAFPs-Mv-BiTCN [32] precisely boosted the prediction of AFP by encompassing Contextual and semantic relations using skip-gram and ProtBERT-BFD and PsePSSM-DWT enriching the evolutionary feature representation, finally, confirming the proposed multi-view deep learning approach attracting framework's strength in the field of bioinformatics. The most widely used algorithm for analyzing any kind of input image is a Convolutional Neural Network (CNN), a subset of deep neural networks used in deep learning. CNN made significant contributions to medical imaging through its application in digital mammography for computer-assisted identification of microcalcifications [33], as well as computer-assisted diagnosis of lung nodules in CT datasets. This is despite the recent revelation of CNN's true power. Recent applications of CNN include the recognition of skin lesions , the segmentation of the pancreas in CT images , the measurement of the carotid intima-media thickness in ultrasound image data , the segmentation of multimodality isointense infant brain images, and the specification of neural membranes in electron microscopy images. A system for categorizing skin lesions was presented by Esteva et al. [34]was using the Inception v3 model. The Inception v3 model that has been used three lesion datasets. This model is able to distinguish between skin lesions that are benign and malignant. In addition, benign seborrheic keratoses and keratinocyte carcinomas can be found on the region of interest by the model. Their analyzed strategy outperformed dermatologists in terms of lesion classification accuracy. In the Dermaquest database, Jafari et al. [35] studied a skin lesion segmentation technique that included deep CNN for picture preprocessing, segmentation, lesion recognition, and classification. With a 95.86% accuracy rate, Namozov and Cho [36] illustrated the function of activation functions on CNN for lesion identification on the ISIC2018 [37] dataset (consisting of 10015 pictures). Using the VGG-SegNet approach to extract information on the ISIC2016 [38,39] dataset with improved values of jaccard-index, dice, and accuracy, Kadry et al. [40] suggested a CNN model to identify skin melanoma from digitized dermoscopy images. For the detection and segmentation of melanoma lesions, Adegun and Viriri [41] studied an improved encoder-decoder network. Using the softmax classifier for pixel-wise classification on the PH2 and ISIC2017 [42] datasets, their system makes use of multi-stage and multi-scale methodologies. Azeem et al. [43] in 2023 investigated on skin cancer images classification with put forwarded SkinLesNet CNN on datasets, HAM10000 and ISIC2017 datasets. His findings were accuracy 90.00%, Precision 89.00%, Recall 87.00% and F1-score 85.00% with Adam optimizer, Relu and "100" epocs . Houssein et al. [44] in 2024 proposed a CNN model to identify skin cancer on data set ISIC-2019 and HAM10000 with traing accuracy 97.1% and 98.8% repectively with high rate of misclassification of class "4" anf "5" in confusion matrix. F1 Score 98.4% , Recall 98.5% and precision 98.5%. For the detection and segmentation of melanoma lesions, The performance problems of the GoogLeNet, AlexNet, ResNet, and VGGNet models were combined to create the ensemble of CNN models by Harangi [45] for melanoma, seborrheic keratosis, and common naevus classification. Four probabilistic fusion procedures are suggested for combining the models. These techniques include, among others, the sum of the maximal probabilities, the product of the probabilities, the total of the odds, and direct plurality voting. The maximum probabilities process number performs better than the other fusion techniques. This investigation shows that joined models outperform solo CNN models in skin lesion recognition.

## 3 Problem definition

Non-linear activation functions, a type of mathematical engine, allow neural networks to learn from complex and unstructured input. They enable the network to learn from the input and provide accurate predictions by introducing non-linearity. Neural networks are not designed to process complex data; otherwise, they would be nothing more than linear regression models without non-linear activation functions. Complex interactions between inputs and outputs cannot be described by linear models; neural networks require non-linearity. In complex and high-dimensional data, it enables the network to learn from the input, adapt to changes, and provide accurate predictions. For further explanation, consider an elementary perceptron model with an input vector $X$, a bias term $b$, and a weight matrix $W$. The output function $Y$'s definition is as follows:

$$Y = W^T X + b \tag{1}$$

These outputs from each layer, which by default are linear, are fed into the following layer in multilayered networks in order to produce the desired output. Each neuron uses a non-linear activation function to the calculated weighted total in order to break away from input-output mappings that are strictly linear. The output of the models is shown below, and the mappings of the equation lead to linear results:

$$Y = \left(w_1 x_1 + w_2 x_2 + w_3 x_3 + \cdots + w_n x_n\right) + b \tag{2}$$

ReLU, which is simple and effective, is one of the most often used activation functions in deep learning. ReLU activation function is in fact nonlinear. Even though it seems to be linear for positive values (output equals input), it really causes nonlinearity by "turning off" for negative inputs. The ability to mimic complex, non-linear interactions in data is a feature of neural networks. $Z =$ ReLU (Y) that was produced non-linearity.

$$Z = \max\left(0, Y\right) = f\left(Y\right) = \begin{cases} 0 & : \ Y < 0 \\ Y & : \ Y \geq 0 \end{cases} \tag{3}$$

ReLU is currently one of the most widely used activation functions in feed-forward neural networks because of their simplicity and properties that help to combat gradients disappearing issue. However, ReLU has also certain disadvantages affecting its performance and training processes. Here are some of the disadvantages of using ReLU: There are various situation that causes incompatibility with batch normalization, missing smoothness, expanding gradient problem, vanishing gradient problem, output saturation and the model is not suitable for all types.

$$Z\left(Y\right) = \begin{cases} Y & if \ Y \geq 0 \\ \beta\left(e^Y - 1\right) & if \ Y < 0 \end{cases} \tag{4}$$

The ELU activation functions have been popular among the deep learning architectures work because of its ability in handling negatives values coupled with the ability to reduce the vanishing gradient problem. However, its computational complexity becomes an issue especially for GPUs and the memory requirements for the models as they scale up due to exponential mathematical computation. Moreover, optimization is also affected in the same way due to the non-monotonic property of ELU that slows down the convergence speed. To

overcome these limitations. The idea of this new function is to keep the positive aspects of ELU and ReLU that were described above, at the same time, improve computational speed and absence of negative activations in the neurons. In this way, the idea of using algebraic functions, monotonic NGNDG-AF is solutions to the optimization processes, training convergence, and memory allocation. Thus, the idea of developing NGNDG-AF can be seen as a breakthrough in the effort to address the problems caused by current activation functions, as well as in enhancing the effectiveness of deep learning models and many applications in various fields .

## 4 Proposed activation function

The generation of non-linear changes in neural networks is generally done by activation functions. In this article, we have introduced a new activation function family named "NGNDG-AF" given in Eq (5), and the facts mentioned below are important to finding the universal approximation ability of deep neural networks with the help of such activation functions and learning higher-order polynomial networks. The enhancement of ReLU and ELU has formed the basis of the current development.

$$f(x) = \begin{cases} g(x) & : \ x \geq 0 \\ h(x) & : \ x < 0 \end{cases} \tag{5}$$

We specially utilized two functions in this study $g(x) = \beta x$ and $h(x) = \frac{\beta x}{1+x^2}$, where $\beta$ is a constant parameter that influences the sign of the output related to negative inputs, and $\beta x$ expression controls the polarity when input is negative value. Specifically, if $\beta$ is positive, negative then $h(x) = \frac{\beta x}{1+x^2}$ is negative , positive respectively.

$$f(x) = \begin{cases} \beta x & : \ x \geq 0 \\ \frac{\beta x}{1+x^2} & : \ x < 0 \end{cases} \tag{6}$$

$$f'(x) = \begin{cases} \beta & : \ x \geq 0 \\ \frac{f(x)}{\beta x} - \frac{2(f(x))^2}{\beta} & : \ x < 0 \end{cases} \tag{7}$$

Fig 1 shows a graph of the NGNDG-AF activation function for both positive and negative values of parameter $\beta$. The function illustrates how NGNDG-AF acts in response to various $\beta$ values, emphasizing the influence on the activation response for both positive and negative inputs. This characteristic underlines its flexibility to a variety of activation patterns, providing a smooth transition from ReLU's characteristics to more complex responses controlled by different values.

Fig 2 displays the activation patterns of the Sigmoid, ReLU, ELU and NGNDG-AF in the network model. NGNDG-AF stands apart from other models despite their similarities by successfully absorbing minute negative inputs, which raises the model's sensitivity. While the unboundedness in the example above speeds up training, the lower boundedness functions as a regularization technique by excluding some inputs from the range of zero to negative infinity. First drawbacks of ELU , it becomes saturated for large negative value so vanished gradient problem create but NGNDG-AF is not saturated for this value. Drawback of ELU and ReLU , $\beta$ in NGNDG-AF is controlling the slop for value greater then zero. Together, these properties make NGNDG-AF a versatile activation function that has the potential to enhance network performance. So on base of above properties of NGNDG-AF ,it has fast convergence that NGNDG-AF demonstrates in Fig 3.

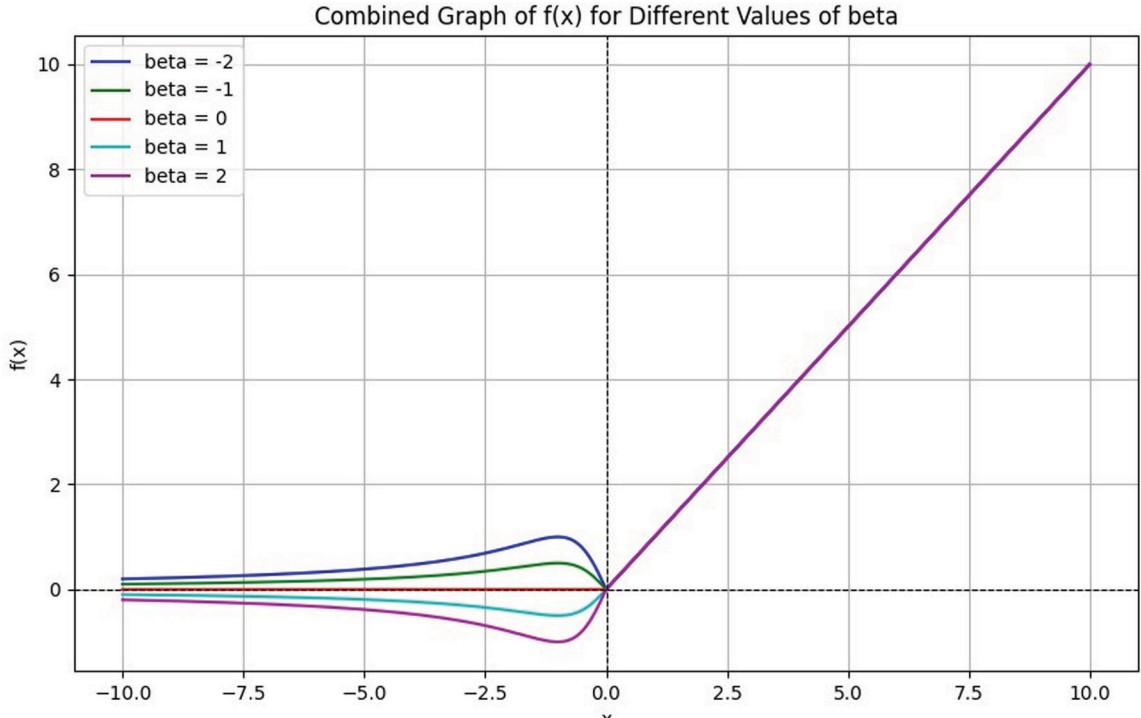

**Fig 1. Variation graph of NGNDG-AF for different value of $\beta$.**

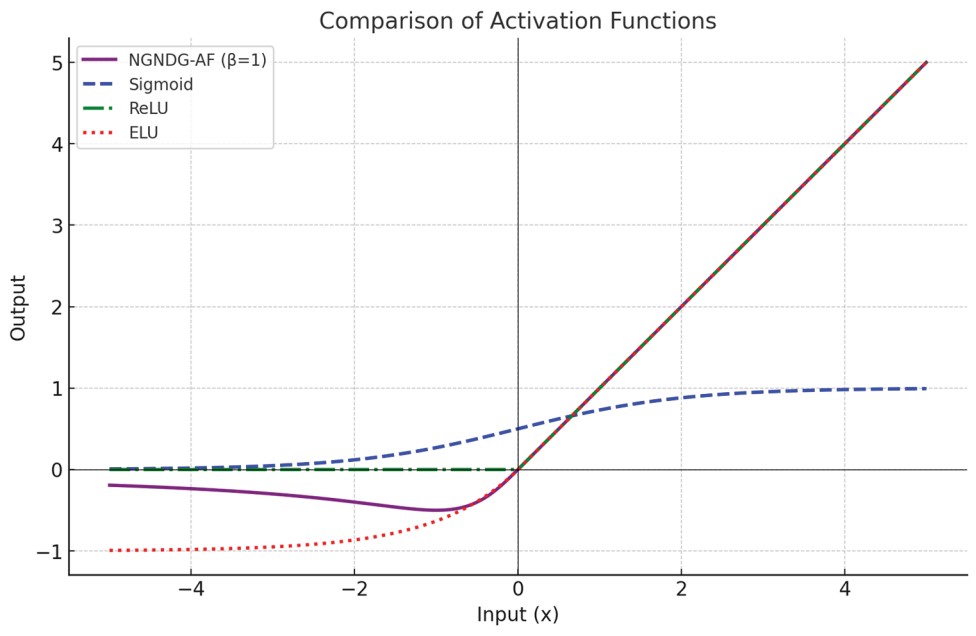

**Fig 2. Comparison of different activation function with NGNDG-AF.**

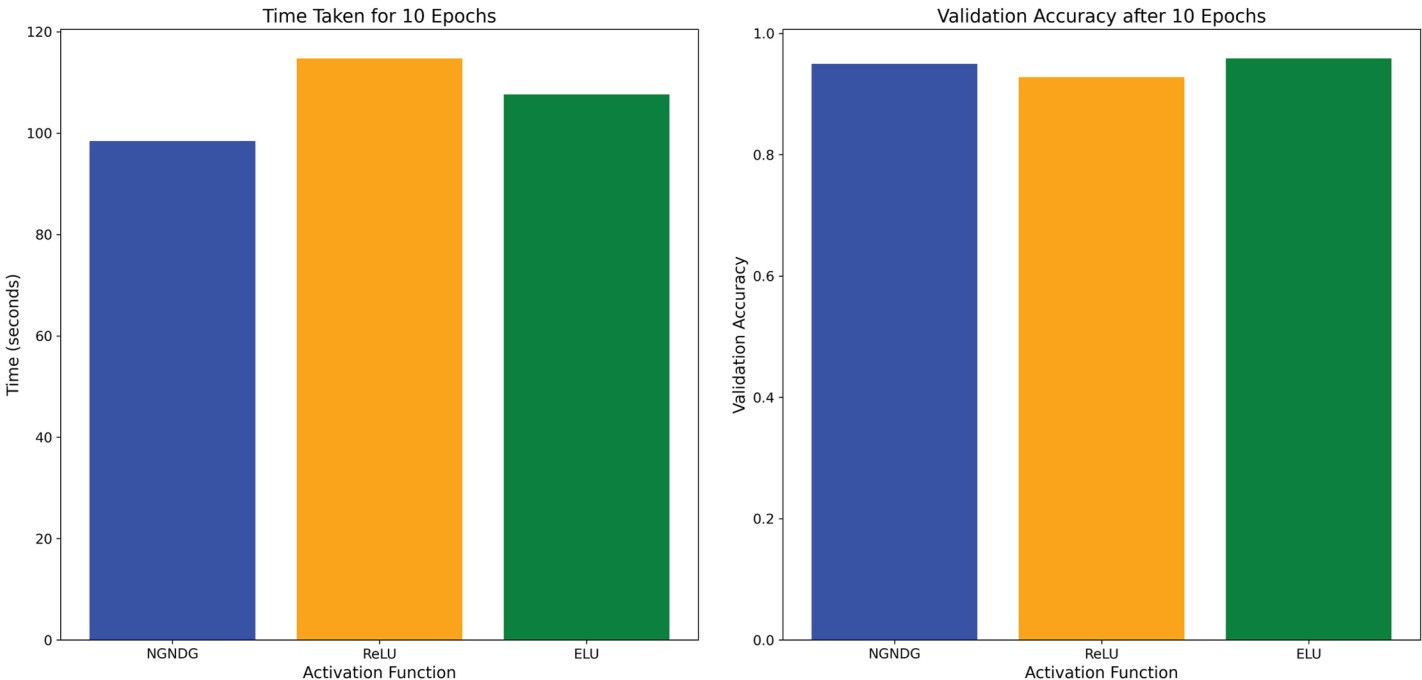

**Fig 3. Performance comparison of NGNDG-AF, ReLU and ELU.**

## 5 Enhanced visualization of CNN: Integrating NGNDG-AF into Grad-CAM

Grad-CAM is a technique used to visualize and understand the regions of an input image that are important for predicting a specific class in a CNN model. It achieves this by analyzing the gradients of the class score with respect to the feature maps of the last convolutional layer. Let's break down the process step by step: In the framework of a CNN model, let's extract the whole mathematical expression generated by Grad-CAM for a NGNDG-AF.

- **Defining the activation function (NGNDG-AF(x)):** The NGNDG-AF(x) activation function is defined.
- **Calculating the gradient:** We want to determine the gradient between the class score $y^c$ and the activations of the k-th channel of the final convolutional layer, denoted as $A^k(i,j)$. The gradient is calculated as: $\frac{\partial y^c}{\partial A^k(i,j)}$. This gradient represents how changes in the activations of the k-th channel affect the class score $y^c$.
- **Finding the relevance weights ($\alpha_k^c$):** The relevance weights, represented by ($\alpha_k^c$), are computed to emphasize the role that each channel plays in forecasting the target class. The following formula is used to calculate these normalized weights $\alpha_k^c = \frac{1}{Z} \sum_i \sum_j \frac{\partial y^c}{A^k(i,j)}$. Whereas the normalizing term is $Z$.
- **Final class discriminate silence map** $L_{Grad-Cam}$: The Grad-CAM score for a specific spatial location $(i,j)$ and channel $k$ is computed as the weighted sum of the feature maps, multiplied by the activation function $NGNDG - AF(x) = f(x)$ specified in Eq (6).

$$L_{Grad-Cam} = f(\sum_K \alpha_k^c A^k(i,j)). \tag{8}$$

- **Completing final class discrimate silence map** $L_{Grad-Cam}$ **with NGNDG-AF**: Finally, if we incorporate the $NGNDG-AF(x) = f(x)$ activation function into the Grad-CAM score, it becomes

$$L_{Grad-Cam} =$$
$$f(\sum_K \alpha_k^c A^k(i,j), \frac{\beta \sum_K \alpha_k^c A^k(i,j)}{1 + (\sum_K \alpha_k^c A^k(i,j))^2}). \tag{9}$$

This equation provides the Grad-CAM score for class C at spatial location (i,j), considering the contribution of each channel weighted by its relevance($\alpha_k^c$) and applying the NGNDG-AF(x) activation function. By analyzing the Grad-CAM scores, we can visualize which regions of the input image are important for the model's prediction of a specific class.

## 6 Enhanced visualization of CNN: Integrating NGNDG-AF into Grad-CAM++

An modification of Grad-CAM called Grad-CAM++ adds positive as well as negative gradients to give a more thorough representation of the relative importance of the various regions in the input image. Grad-CAM++ is a mathematical formulation that builds upon the Grad-CAM formulation. We are going to refer to the NGNDG-AF and find the Grad-CAM++ formulation We can reformulate the structure of the weights $\omega_k^c$ to fix this problem by taking a weighted average of the pixel-wise gradients. The new structure of the weights can be represented as follows and $f(\frac{\partial Y^c}{\partial A_{ij}^k})$ is defined in Eq (6).

- **Finding the relevance weights** $\alpha_k^c$:

$$\omega_k^c = \sum_i^N \sum_J^N \alpha_{ij}^{kc} f(\frac{\partial Y^c}{\partial A_{ij}^k}) \tag{10}$$

feature map $Y^c$ is defined as

$$Y^c = \sum_k^N \{\sum_a^N \sum_b^N \alpha_{ab}^{kc} f(\frac{\partial Y^c}{\partial A_{ab}^k})\}[\sum_i^N \sum_j^N A_{ij}^k] \tag{11}$$

calculating derivatives of $Y^c$ in relation with $A_{ij}^k$ then

$$\alpha_{ij}^{kc} = \frac{\frac{\partial^2 Y^c}{(\partial A_{ij}^k)^2}}{2f'(\frac{\partial Y^c}{\partial A_{ab}^k}) + \sum_a^N \sum_b^N A_{ab}^k \left\{ \begin{array}{l} f''(\frac{\partial Y^c}{\partial A_{ij}^k})\frac{\partial^2 Y^c}{(\partial A_{ij}^k)^2} \\ +f'(\frac{\partial Y^c}{\partial A_{ij}^k})\frac{\partial^3 Y^c}{(\partial A_{ij}^k)^3} \end{array} \right\}} \tag{12}$$

- **Calculating derivatives of** $Y^c$ **in relation with** $A_{ij}^k$: Grad-CAM++ introduces a weighted sum of the second-order gradients, expanding upon Grad-CAM's focus on the gradient information flowing into the final convolutional layer. By incorporating this weighted sum,

Grad-CAM++ not only evaluates the importance of the gradient but also considers its interaction with the surrounding context.

The raw scores (often referred to as logits) for each class are transformed into a probability distribution using the softmax function. This function assigns probabilities to each class based on the input scores. Here $Y^c$ represents the raw score (logit) for the $c$-th class, and the softmax function assigns probabilities to each class based on these scores $f(x)$ is defined in Eq (6).

$$Y^c = \frac{e^{f(A_{ij}^k)}}{\sum_k e^{f^k(A_{ij}^k)}} \tag{13}$$

In the penultimate layer $f(A_{ij}^k)$ is the score pertaining to output class $k$ where the index $k$ runs over all output classes. The first derivative

$$\frac{\partial Y^c}{\partial A_{ij}^k} = Y^c \left[ \frac{\partial f^c(A_{ij}^k)}{\partial A_{ij}^k} - \sum_k Y^k \frac{\partial f^k(A_{ij}^k)}{\partial A_{ij}^k} \right] \tag{14}$$

In neural networks employing NGNDG activation functions, where the second derivative is null for positive values and non-null for negative ones, the inclination is typically toward considering it as non-zero.

$$\frac{\partial^2 Y^c}{\left(\partial A_{ij}^k\right)^2} = \frac{\partial Y^c}{\partial A_{ij}^k} \left[ \frac{\partial f^c(A_{ij}^k)}{\partial A_{ij}^k} - \sum_k Y^k \frac{\partial f^k(A_{ij}^k)}{\partial A_{ij}^k} \right]$$
$$+ Y^c \left[ \frac{\partial^2 f^c(A_{ij}^k)}{\left(\partial A_{ij}^k\right)^2} - \sum_k \left[ \frac{\partial Y^k}{\partial A_{ij}^k} \frac{\partial f^k(A_{ij}^k)}{\partial A_{ij}^k} + Y^k \frac{\partial^2 f^k(A_{ij}^k)}{\left(\partial A_{ij}^k\right)^2} \right] \right] \tag{15}$$

and

$$\frac{\partial^3 Y^c}{\left(\partial A_{ij}^k\right)^3} = \frac{\partial^2 Y^c}{\left(\partial A_{ij}^k\right)^2} \left[ \frac{\partial f^c(A_{ij}^k)}{\partial A_{ij}^k} - \sum_k Y^k \frac{\partial f^k(A_{ij}^k)}{\partial A_{ij}^k} \right]$$
$$+ 2 \frac{\partial Y^c}{\left(\partial A_{ij}^k\right)} \left[ \frac{\partial^2 f^c(A_{ij}^k)}{\left(\partial A_{ij}^k\right)^2} - \sum_k \left[ \frac{\partial Y^k}{\partial A_{ij}^k} \frac{\partial f^k(A_{ij}^k)}{\partial A_{ij}^k} + Y^k \frac{\partial^2 f^k(A_{ij}^k)}{\left(\partial A_{ij}^k\right)^2} \right] \right]$$
$$+ Y^c \left[ \left[ \frac{\partial^3 f^c(A_{ij}^k)}{\left(\partial A_{ij}^k\right)^3} - \sum_k \left[ \frac{\partial^2 Y^k}{\left(\partial A_{ij}^k\right)^2} \frac{\partial f^k(A_{ij}^k)}{\partial A_{ij}^k} + \right. \right. \right.$$
$$\left. \left. \left. 2 \frac{\partial Y^k}{\partial A_{ij}^k} \frac{\partial^2 f^k(A_{ij}^k)}{\left(\partial A_{ij}^k\right)^2} + Y^k \frac{\partial^3 f^k(A_{ij}^k)}{\left(\partial A_{ij}^k\right)^3} \right] \right] \right] \tag{16}$$

- **Final class discriminate silence map** $L_{Grad-Cam++}$: The gradient weights can be derived by substituting Eq (15) and Eq (16) into Eq (12). The integration of NGNDG-AF into Grad Cam++ is generalized form of integration Relu in Grad Cam++.

   Final class discriminate silence map $L_{Grad-Cam++}$ is

$$L_{Grad-Cam++} = f\left(\sum_k \omega_k^c A_{ij}^k\right) \tag{17}$$

from (6), (10) and (12)

$$L_{Grad-Cam++} = f\left( \sum_k \left[ \sum_i^N \sum_j^N \alpha_{ij}^{kc} f\left( \frac{\partial Y^c}{\partial A_{ij}^k} \right) \right] A_{ij}^k \right) \qquad (18)$$

The discriminate silence map is a final class specifically designed for implementing the Grad-CAM++ algorithm, a sophisticated technique in deep learning used for visualizing significant regions within input data. Unlike its predecessors, Grad-CAM++ with NGNDG-AF integrates both positive and negative gradients, resulting in a refined heatmap. This heatmap not only highlights relevant features but also identifies neglected areas by the model. This class undertakes the task of computing gradients, merging them, and subsequently generating the heatmap. As a result, it offers valuable insights into where the neural network directs its attention and where it remains indifferent. By employing visualization techniques, it facilitates the interpretation of the model's behavior, thereby becoming an indispensable tool for understanding the inner workings of CNN models.

## 7 Model of optimized convolutional neural network

Number of filters in each layer in normal and conventional CNN architectures. Besides, the activation layers for NGNDG-AF and the particular distribution of the convolutional blocks make the CNN design unique from the conventional ones. This is due to the opportunity to use multiple branches within the convolutional blocks, which in its turn provides more complex feature extraction from the given images. Further, they introduced the concept of batch normalization layer to stabilise and improve the efficiency of the training through normalising the output of activation functions between layers. Another type of hidden layers is dropout layers for the purpose of preventing over fitting during a training. the certain ratio of nodes are dropping during a training. Thus, architecture of the offered CNN reflects the depth, complexity and specific structure of layers considered being appropriate for the feature extraction and images categorization. Let's break down the key characteristics mentioned in my description:

- **Number of layers**: In more detail, the CNN architecture is comprised of 34 layers.
- **Primary blocks**: The architecture is made up of four main components. The first block situated at the bottom of the network is made up of one convolution layer. The representational form of these is the second block with three branches, the third block, and fourth block with each having three branches.
- **Layers in branches**: Each branch in the second, third, and fourth blocks includes a combination of layers: batch normalization, maximum pooling, and an activation layer called NGNDG-AF.
- **Convolution layers in branches**: The convolutional layers in the branches vary. The first branch has a single $1 \times 1$ Conv2D layer. The second branch has two $3 \times 3$ Conv2D layers. The third branch has two $3 \times 3$ and one $1 \times 1$ Conv2D layers.
- **Fully connected layers**: The first fully connected layer consists of 256 nodes and includes NGNDG-AF activation and dropout. The second hidden layer has 64 nodes, following the same construction as the first hidden layer.
- **Classification layer**: The last layer is the classification layer with 7 nodes, defined by the SoftMax activation function.

    This CNN architecture differs from existing designs in several aspects, including the use of unique convolution layers, filter numbers, filter sizes, and fully connected layers. Notably,

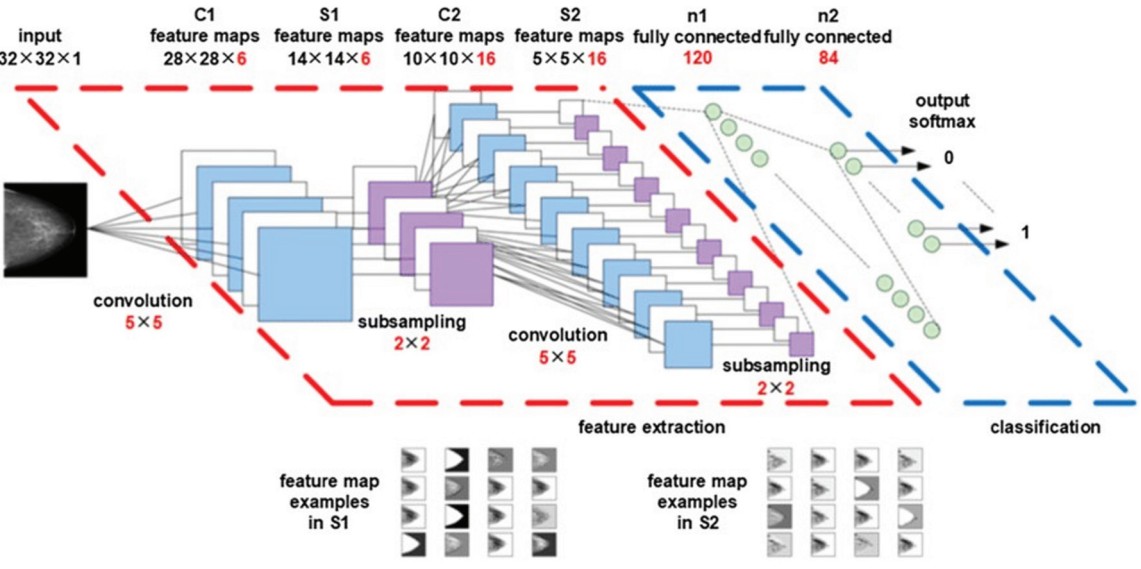

**Fig 4. Architecture of deep convolutional neural network.**

it follows a distinct pattern opposed to the conventional approach of progressively increasing filter sizes and layers. Overall, described CNN architecture appears to be a sophisticated design aimed at automatic feature extraction and image categorization, leveraging unique characteristics in its structure and layers.

## 8 Spare categorical cross entropy Loss and Adam optimization

When target labels for a multi-class classification are integers, like class indices, the sparse categorical crossentropy loss function is frequently used. It's calculated by determining the cross-entropy between the actual probability distribution (the ground truth labels), which is represented by the symbol $y$, and the predicted probability distribution that the model produces, which is represented by the symbol p. Whereas p is the projected probability vector for each class produced by the softmax activation function, $y$ is usually a one-hot encoded vector that represents the true class for each sample. This loss function is expressed mathematically and is obtained from the cross-entropy between both distributions.

- **Sparse categorical crossentropy loss**: Sparse Categorical Crossentropy Loss is defined as

$$L = \frac{-\sum_{i=1}^{N} 1 \sum_{j=1}^{C} y_{i,j} \log(p_{i,j})}{N} \tag{19}$$

from (6) and $p_{i,j}$ defined as

$$p_{i,j} = Soft\max(f(x_{i,j})) \tag{20}$$

$$\frac{dp_{i,j}}{df(x_{i,j})} = p_{i,j}(1 - p_{i,j})$$

Where $N$ is the batch's total number of samples. $C$ is how many classes there are. $y_{i,j}$ Is the actual probability that example $i$ is a member of class $j$. When $j$ is the true class, this is 1, and when it's not, it's 0. .

- **Derivative calculation in cross-entropy loss for multiclass classification**: We want to find the derivative $\frac{dL}{dp_{i,j}}$, which represents how the loss changes with respect to the predicted probability $p_{i,j}$. We start by applying the chain rule.

$$\frac{dL}{dp_{i,j}} = -\frac{1}{N} \sum_{i=1}^{N} \sum_{j=1}^{C} \frac{d}{dp_{i,j}} (y_{i,j} \log(p_{i,j})) \tag{21}$$

If $j$ is the correct class then $y_{i,j} = 1$ and if not correct class $y_{i,j} = 0$ substitute these back into the derivative expression (21).

$$\frac{dL}{dp_{i,j}} = -\frac{1}{N} \sum_{i=1}^{N} \frac{1}{p_{i,j}}. \tag{22}$$

This expression gives us the derivative of the loss function with respect to the predicted probability $p_{i,j}$. It tells us how much the loss changes when we change the predicted probability of the correct class for each sample. This gradient is crucial for updating the model parameters.

- **Gradient calculation in neural networks with NGNGD-AF**: To determine the gradient of the NGNGD-AF's output $f(x_{i,j})$ in relation to the weighted sum $x_{i,j} = w_{i,k} z_{i,j} + b_j$, we need to compute the derivative of the NGNDG-AF function with respect to $x_{i,j}$ . We have already been given the expression Eq (6) and Eq (7) for the NGNGD-AF function and it's derivative. we utilized the chain rule to determine the loss's gradient in relation to the model's parameters.

$$\frac{\partial L}{\partial w_{i,j}} = \frac{\partial L}{\partial p_{i,j}} \frac{\partial p_{i,j}}{\partial f(x_{i,j})} f'(x_{i,j}) \frac{\partial x_{i,j}}{\partial w_{i,k}} \tag{23}$$

$$\frac{\partial L}{\partial b_j} = \frac{\partial L}{\partial p_{i,j}} \frac{\partial p_{i,j}}{\partial f(x_{i,j})} f'(x_{i,j}) \frac{\partial x_{i,j}}{\partial b_j} \tag{24}$$

the gradient of the weighted sum $x_{i,j} = w_{i,k} z_{i,j} + b_j$ in relation to the model's parameters $w_{i,k}$ and $b_j$ are $z_{i,j}$ respectively and from Eq (7), Eq (20), Eq (21), Eq (22), Eq (23), and Eq (24)

$$\frac{\partial L}{\partial w_{i,j}} = z_i (p^2{}_{i,j} - p_{i,j}) \left( \frac{\sum_{i=1}^{N} \frac{1}{p_{i,j}}}{N} \right) \begin{cases} \beta & : \ 0 \le x_{i,j} \\ \frac{\beta(1-x_{i,j}^2)}{(1-x_{i,j}^2)^2} & : \ x_{i,j} < 0 \end{cases} \tag{25}$$

$$\frac{\partial L}{\partial b_j} = (p^2{}_{i,j} - p_{i,j}) \left( \frac{\sum_{i=1}^{N} \frac{1}{p_{i,j}}}{N} \right) \begin{cases} \beta & : \ 0 \le x_{i,j} \\ \frac{\beta(1-x_{i,j}^2)}{(1-x_{i,j}^2)^2} & : \ x_{i,j} < 0 \end{cases} \tag{26}$$

- **Optimizing neural networks with Adam algorithm**: The Adam algorithm is a popular optimization technique used to update model parameters during the training process. It combines ideas from both momentum and RMSprop algorithms to achieve better convergence properties. Initialize $m_0$ and $v_0$ to zeros or other suitable initial values and Set hyperparameters $\alpha$ learning rate, $\beta_1, \beta_2$ and $\varepsilon$. Update the moving averages of the gradients

squared

$$m_t = \beta_1 m_{t-1} + (1 - \beta_1)\frac{\partial L}{\partial w_{i,j}} \tag{27}$$

$$v_t = \beta_2 m_{t-1} + (1 - \beta_2)\left(\frac{\partial L}{\partial w_{i,j}}\right)^2 \tag{28}$$

Correct bias in the moving averages.

$$\overset{\vee}{m}_t = \frac{m_t}{1 - \beta^t_1} \tag{29}$$

$$\overset{\vee}{v}_t = \frac{v_t}{1 - \beta^t_2} \tag{30}$$

Update the parameters using the moving averages and the learning rate

$$w_{i,k} = w_{i,k} - \alpha\frac{\overset{\vee}{m}_t}{\sqrt{\overset{\vee}{v}_t} + \varepsilon} \tag{31}$$

$$b_j = b_j - \alpha\frac{\overset{\vee}{m}_t}{\sqrt{\overset{\vee}{v}_t} + \varepsilon} \tag{32}$$

Continue the same procedure until convergence, or for a predetermined number of epochs. The steps to calculate the gradient of the sparse categorical cross entropy loss with respect to the predicted probabilities are described in this procedure, along with an application to train a neural network using the Adam optimizer.

## 9 Evaluation of performance and experimental results

NGNDG-AF(an algebraically-based activation function) was combined with optimized CNN, ResNet152V2, ADAM and RMSPROP optimizers to form a system that tackled challenging tasks such as skin cancer image classification of HAM10000. There were significant improvements witnessed by this integration across diverse performance metrics. In addition, incorporating NGNDG-AF into the optimized CNN, ResNet152V2 in conjunction with ADAM and RMSPROP optimizers resulted in improved accuracy, F1 score, AUC, precision, true positive, true negative, reduction in computation cost as well as accelerated training time due to faster convergence speed and other important evaluation criteria. Furthermore, the interpretation and visualization of Grad-CAMs and Grad-CAM++ were improved by synergistic requirements of NGNDG-AF which provided more insights on how it works. These are encouraging first results for deepening our understanding of the impact on different metrics or implications from applying advanced visualization to understand model behavior better when using NGNDG-AF activation functions.

In provided Table 1, Figs 5, 6 and Figs 8, 9, 10 two different optimization algorithms, Adam and Rmsprop, were compared for performance on two different Convolutional Neural Network architectures: the optimized CNN and ResNet152V2. Both these architectures employed the NGNDG-AF activation function. Let's first look at the findings of this study with regards to the results obtained when using ResNet152V2 architecture. In whatever optimizer was used, be it Adam or Rmsprop; ResNet152V2 has shown to have an exceptional accuracy rate since the scores for validation accuracy range above 97.7%. This means that

**Table 1. Performance metrics for HAM10000 data.**

| Model (NGNDG-AF) | Accuracy (Train) | Accuracy (Valid) | Loss (Train) | Loss (Valid) |
|---|---|---|---|---|
| Optimized CNN (Adam) | 0.9935 | 0.9857 | 0.0640 | 0.0224 |
| Optimized CNN (Rmsp) | 0.9956 | 0.9788 | 0.0155 | 0.1877 |
| ResNet152V2 (Adam) | 0.9979 | 0.9752 | 0.0072 | 0.1523 |
| ResNet152V2 (Rmsp) | 0.9980 | 0.9736 | 0.0089 | 0.2639 |

It is a table that presents training and validation accuracy and loss for different models developed with NGNDG-AF focusing on the HAM10000 dataset.

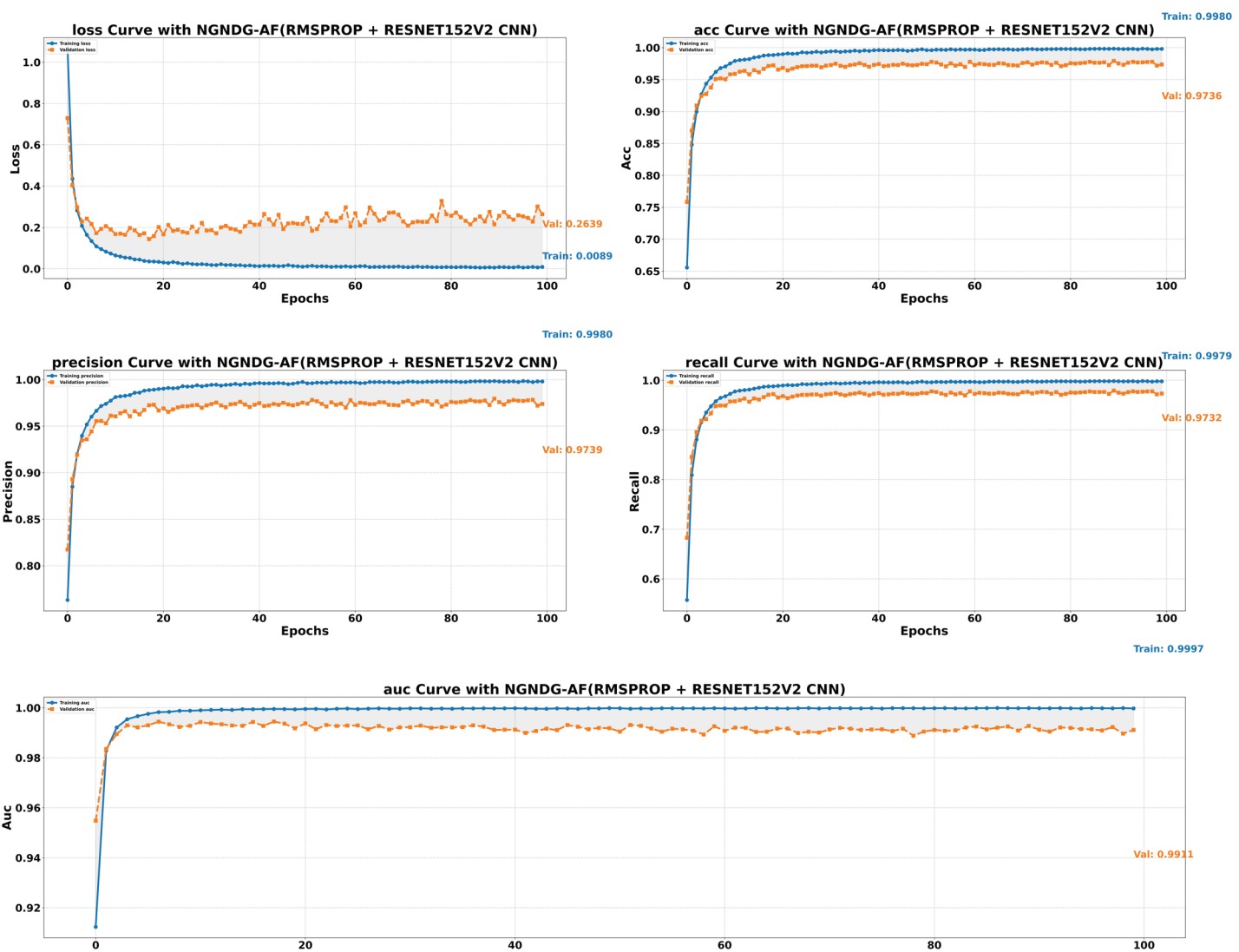

**Fig 5. Performance metric of ResNet152V2 CNN with Ramsprop for $\beta = 1$.**

ResNet152V2 can recognize images as highlighted by the Adam(97. 52%) and Rmsprop (97. 36%) validation accuracy respectively. From this representation, it can be deduced that ResNet152V2 is not only robust but also reliable in tasks of image classification. The

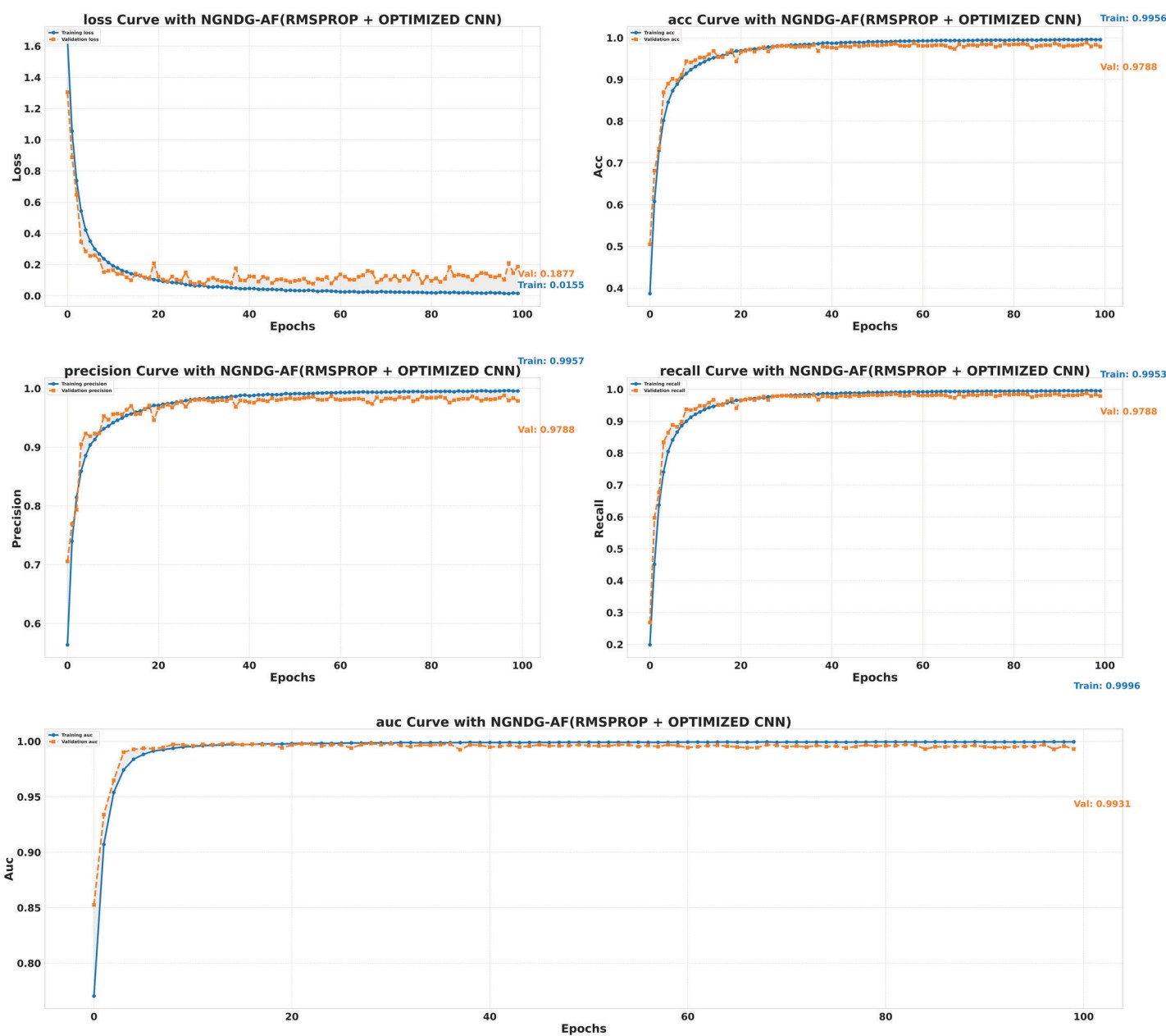

**Fig 6. Performance metric of optimized CNN with Ramsprop for $\beta = 1$.**

optimized CNN architecture was even higher with validation accuracy scores compared to those of ResNet152V2 when discussed now. The optimized CNN was able to get a validation accuracy of 98.57%. While using Adam optimizer it was only able to reach. however using Rmsprop it was able to get 97.88%. Notably, compared to both Optimization algorithm used here, The optimised CNN outperforms resnet152v2 in validation accuracy values also as can seen from the following figures under both optimization algorithms used here. Both, of these model architectures show the high performance that could be attributed to the NGNDG-AF activation function that has been used in both models. Consequently, NGNDG-AF facilitated

better gradient flow during training which made it easier for the model to converge faster and avoid vanishing gradients or exploding gradients issues. Also, NGNDG-AF was instrumental in capturing complex features within images that resulted in improved performance in image classification tasks. The reason behind high validation accuracy of optimized CNN as compared to ResNet152V2 maybe due to its special architecture designed specifically for dataset or task under consideration. Inclusion of multiple branches within convolutional blocks in the architecture of the optimized CNN makes feature extraction more complicated. This enabled the model to pick up very fine details on pictures which contribute towards a higher value of validation accuracy. However, both architectures especially the optimized CNN achieved high rates of validation accuracy that indicated that they were well-designed networks with regard not only o an integrated multinomial but also a group-norm distribution-based activation function and their respective structures involved many hidden layers that allowed them to extract many intricate features from images respectively.. It is notable that this research paper shows how important it is to come up with optimal designs so as to achieve higher degrees of precision when performing image recognition using tailored architectures such as those used in optimized CNNs than what can be obtained through ResNet variants like ResNet152V2.

The Tables 2, 3, and 4, Figs 5, 6, 7, 8, and Figs 13, 14 provided overall performance data for precision, recall, F1-score, support, and AUC for each class in the optimized CNN and ResNet152V2 models using specific optimization techniques (adam and rmsprop), all with the NGNDG-AF activation feature. Analyzing the results for each class revealed consistent high performance across both approaches and optimization techniques. Analyzing the results for each class revealed constant high overall performance across all models and optimization techniques. While discussing the results, it is essential to emphasize the uniformly high performance of the models that is seen in the precision, recall, and F1-score for every class, In addition, the optimized CNN had better performance metrics such as precision (98.59%),

**Table 2. Identification efficiency index for HAM10000 data.**

| Model/Metric | Precision (Train) | Precision (Valid) | Recall (Train) | Recall (Valid) | AUC (Train) | AUC (Valid) |
|---|---|---|---|---|---|---|
| **Optimized CNN (Adam)** | 0.9939 | 0.9859 | 0.9934 | 0.9856 | 0.9996 | 0.9979 |
| **Optimized CNN (Rmsp)** | 0.9957 | 0.9788 | 0.9953 | 0.9788 | 0.9996 | 0.9831 |
| **ResNet152V2 (Adam)** | 0.9981 | 0.9758 | 0.9979 | 0.9748 | 0.9999 | 0.9941 |
| **ResNet152V2 (Rmsp)** | 0.9980 | 0.9739 | 0.9979 | 0.9732 | 0.9997 | 0.9911 |

Efficiency identification metrics index of HAM10000, at precision, recall and AUC for the training and validation processes Table 2.

**Table 3. Performance metrics for HAM10000 dataset.**

| Metric / Class | Class 0 | Class 1 | Class 2 | Class 3 | Class 4 | Class 5 | Class 6 |
|---|---|---|---|---|---|---|---|
| **Precision (Optimized CNN)** | 1.00 | 0.99 | 0.97 | 1.00 | 0.99 | 1.00 | 0.94 |
| **Precision (ResNet152V2)** | 0.99 | 0.98 | 0.94 | 1.00 | 0.99 | 0.99 | 0.91 |
| **Recall (Optimized CNN)** | 1.00 | 1.00 | 1.00 | 1.00 | 0.90 | 1.00 | 0.99 |
| **Recall (ResNet152V2)** | 1.00 | 0.99 | 0.99 | 1.00 | 0.82 | 1.00 | 0.99 |
| **F1 Score (Optimized CNN)** | 1.00 | 1.00 | 0.98 | 1.00 | 0.94 | 1.00 | 0.97 |
| **F1 Score (ResNet152V2)** | 1.00 | 0.99 | 0.97 | 1.00 | 0.89 | 0.99 | 0.95 |
| **AUC (Optimized CNN)** | 1.00 | 0.999996 | 0.999474 | 1.00 | 0.996622 | 1.00 | 0.998994 |
| **AUC (ResNet152V2)** | 0.999997 | 0.999913 | 0.999620 | 1.00 | 0.995963 | 1.00 | 0.998607 |

Shown here are the results of Optimized CNN and ResNet152V2 with activation function being NGNDG-AF and optimizer being Adam.

**Table 4. Performance metrics for HAM10000 dataset.**

| Metric / Class | Class 0 | Class 1 | Class 2 | Class 3 | Class 4 | Class 5 | Class 6 |
|---|---|---|---|---|---|---|---|
| **Precision (Optimized CNN)** | 1.00 | 0.99 | 0.97 | 1.00 | 0.99 | 1.00 | 0.97 |
| **Precision (ResNet152V2)** | 0.99 | 0.98 | 0.96 | 1.00 | 0.99 | 0.99 | 0.94 |
| **Recall (Optimized CNN)** | 1.00 | 1.00 | 1.00 | 1.00 | 0.92 | 1.00 | 0.99 |
| **Recall (ResNet152V2)** | 1.00 | 0.99 | 0.99 | 1.00 | 0.86 | 1.00 | 0.99 |
| **F1 Score (Optimized CNN)** | 1.00 | 0.99 | 0.98 | 1.00 | 0.92 | 1.00 | 0.97 |
| **F1 Score (ResNet152V2)** | 1.00 | 0.99 | 0.98 | 1.00 | 0.92 | 0.99 | 0.97 |
| **AUC (Optimized CNN)** | 0.999938 | 0.999615 | 0.998654 | 0.999938 | 0.995998 | 0.999322 | 0.999322 |
| **AUC (ResNet152V2)** | 0.999999 | 0.999924 | 0.999233 | 1.00 | 0.992463 | 0.998900 | 0.998900 |

The following tables summarize the results of the proposed Optimized CNN and ResNet152V 2 models for the NGNDG-AF activation function and Rmsprop optimizer.

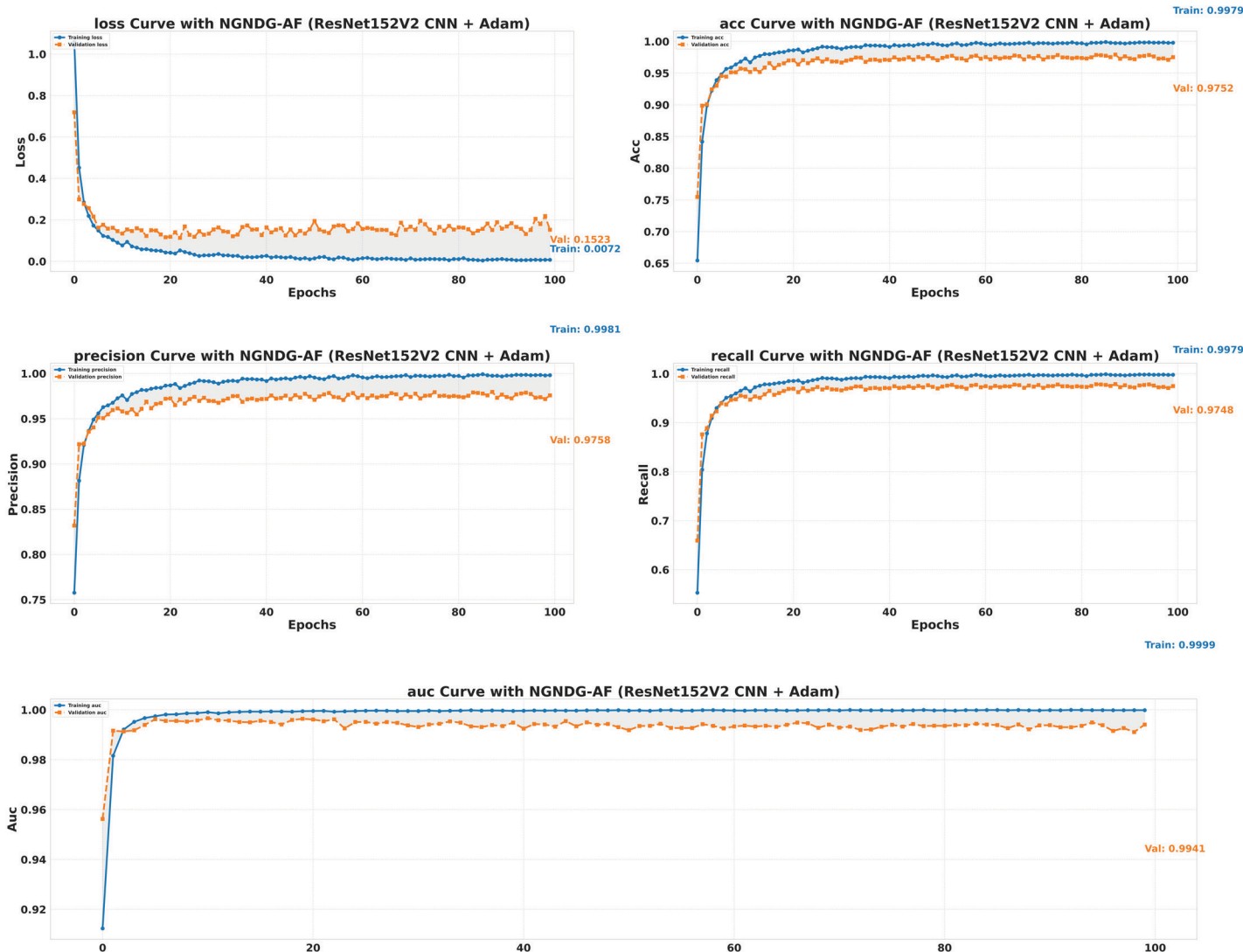

**Fig 7. Performance metric of ResNet152V2 CNN with Adam for $\beta = 1$.**

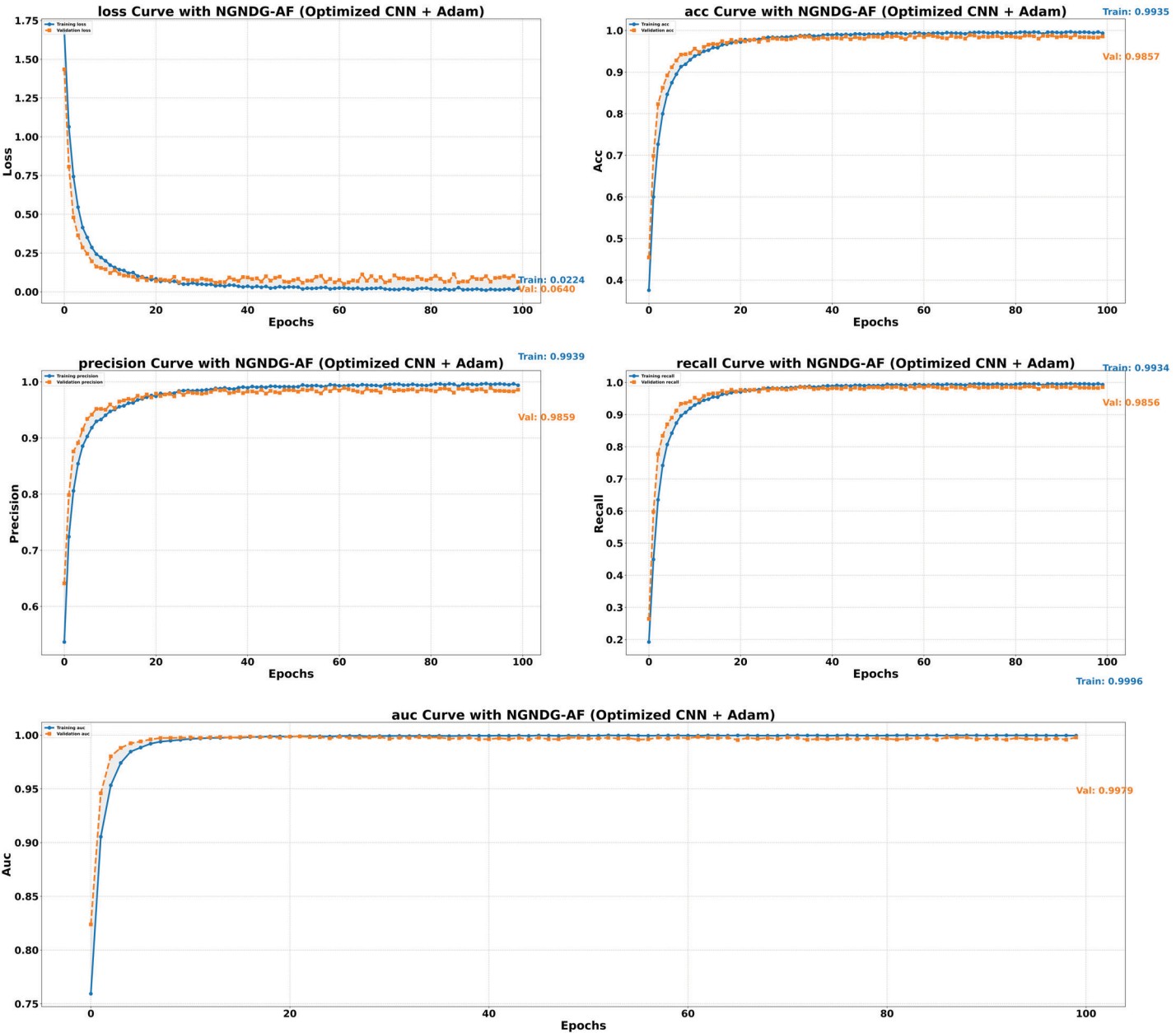

**Fig 8. Performance metric of optimized CNN with Adam for $\beta = 1$.**

recall (98.56%), and AUC (99.79%). Studying classifying CNN system with high value of precision is classifying images correctly labeled with few false positive. This high value of recall ensures that most suitable images are picked. while a high AUC demonstrates the model's strong ability to distinguish between different image classes across thresholds. This implies that it is more effective in classifying skin-lesions accurately, which suggested their ability to correctly classify instances within all classes. Moreover, with AUC scores approximating 1, it indicated the models' accurate identification of positive cases and a lack of false positives among negative cases. With respect to individual classes, most of them attain nearly

perfect precision, recall, and F1-score, thereby implying their capacity in discriminating between different categories without much confusion. A careful analysis of the outcome metrics from this study reveals small performance variances from class to class in terms of precision, recall, and F1-score measures. It is highly likely that the NGNDG-AF activation function also significantly contributes to the success rate of these models as a whole. This behavior smoothens out the training process, ensuring faster convergence on complex data patterns. These characteristics make the models reliable by exhibiting high precision and recall figures on various classes, thus showing their overall effectiveness. The CNN is more efficient and in terms of performance, it is higher than ResNet152V2, especially with regard to precision, recall, and AUC. It can be understood that this improvement results from the specific architecture of CNN optimized for given tasks. This has also been facilitated by the use of NGNDG-AF along with optimization algorithms that have further enhanced the performance of CNN – showing it as a definitive choice for this particular classification task.

The Table 5 included different models' computational costs including ResNet50, Vgg16, Xception, ResNet152v2 and optimized CNN models with various optimization algorithms and activation functions were examined. This review included variables like ram consumption, epoch length, gpu memory usage and total training time. From the findings it was noted that there were huge variations in computational costs depending on the model and settings. Compared to other configurations Adam optimizer plus ReLu function on ResNet50 require fewer gpu and ram resources demonstrating lower computational requirements comparatively.

It is worth mentioning that it took quite some time during the training, but the CNN architecture equipped with NGNDG-AF activation function performed better than any other architectures. This is due to the algebraic base architectures that the NGNDG-AF function enables, where such an activation ensures a seamless flow of gradient and thus enhancing faster convergence during training. The optimized CNN demonstrated in Fig 3 quite remarkable outcomes in relation to training time, which suggested that its construction and selection of an activation function were very beneficial for fast learning. The NGNDG-AF activation function is likely a key factor contributing to the superior performance observed in the optimized CNN, particularly concerning computational costs. For instance, NGNDG-AF has a perk in that it accelerates the convergence process, thereby decreasing the entire time for training. The algebraic nature makes NGNDG-AF superior to ReLU or ELU as well, and such a characteristic is highly valuable to achieve high performance across intricate neural network architectures. It is also shown from the table that CNN models with NGNDG-AF have faster training times, as this model takes very little time per epoch. It turned out that a shorter training duration provided an opportunity to achieve accuracy faster because the CNN network converged at an accelerated rate. The optimized CNN is significantly faster and more efficient as it has a short training time and used few resources. It is, therefore, appropriate to refer to

**Table 5. Computational costs.**

| Metric / Model | ResNet50/VGG16 [46] | ResNet50/Xception [46] | ResNet152V2 | Optimized CNN |
|---|---|---|---|---|
| Optimizer (AF) | Adam (ReLU) | Adam (ReLU) | Adam (NGNDG) | Adam (NGNDG) |
| RAM (GB) | 2.24 | 3.07 | 8.2 / 12.7 | 5.3 / 12.7 |
| GPU (GB) | 9.60 | 14.24 | 8.8 / 15.0 (T4) | 2.8 / 15.0 (T4) |
| Time (Epoch) | 1432 | 1956 | 28.84 | 15 |
| Training Time (s) | 143200 | 195600 | 2884 | 1500 |

Memory consumption/ RAM usage/ GPU requirement/ time to train a model, the number of epochs required for each of the models and optimizers on the given HAM10000 dataset.

**Table 6. Identification efficiency index for HAM10000 dataset**

| Model / Metric | TP | | TN | | FP | | FN | |
|---|---|---|---|---|---|---|---|---|
| | Train | Valid | Train | Valid | Train | Valid | Train | Valid |
| **Optimized CNN (Adam)** | 37,300 | 9,252 | 225,060 | 56,190 | 238 | 132 | 246 | 350 |
| **Optimized CNN (Rmsprop)** | 37,131 | 9,099 | 224,899 | 56,038 | 389 | 284 | 117 | 288 |
| **ResNet152V2 (Adam)** | 37,469 | 9,150 | 225,216 | 56,095 | 72 | 227 | 79 | 237 |
| **ResNet152V2 (Rmsprop)** | 37,470 | 9,135 | 225,213 | 56,077 | 75 | 245 | 78 | 252 |

The results also include True Positives (TP), True Negatives (TN), False Positives (FP), False Negatives (FN) on both the training and the validation data.

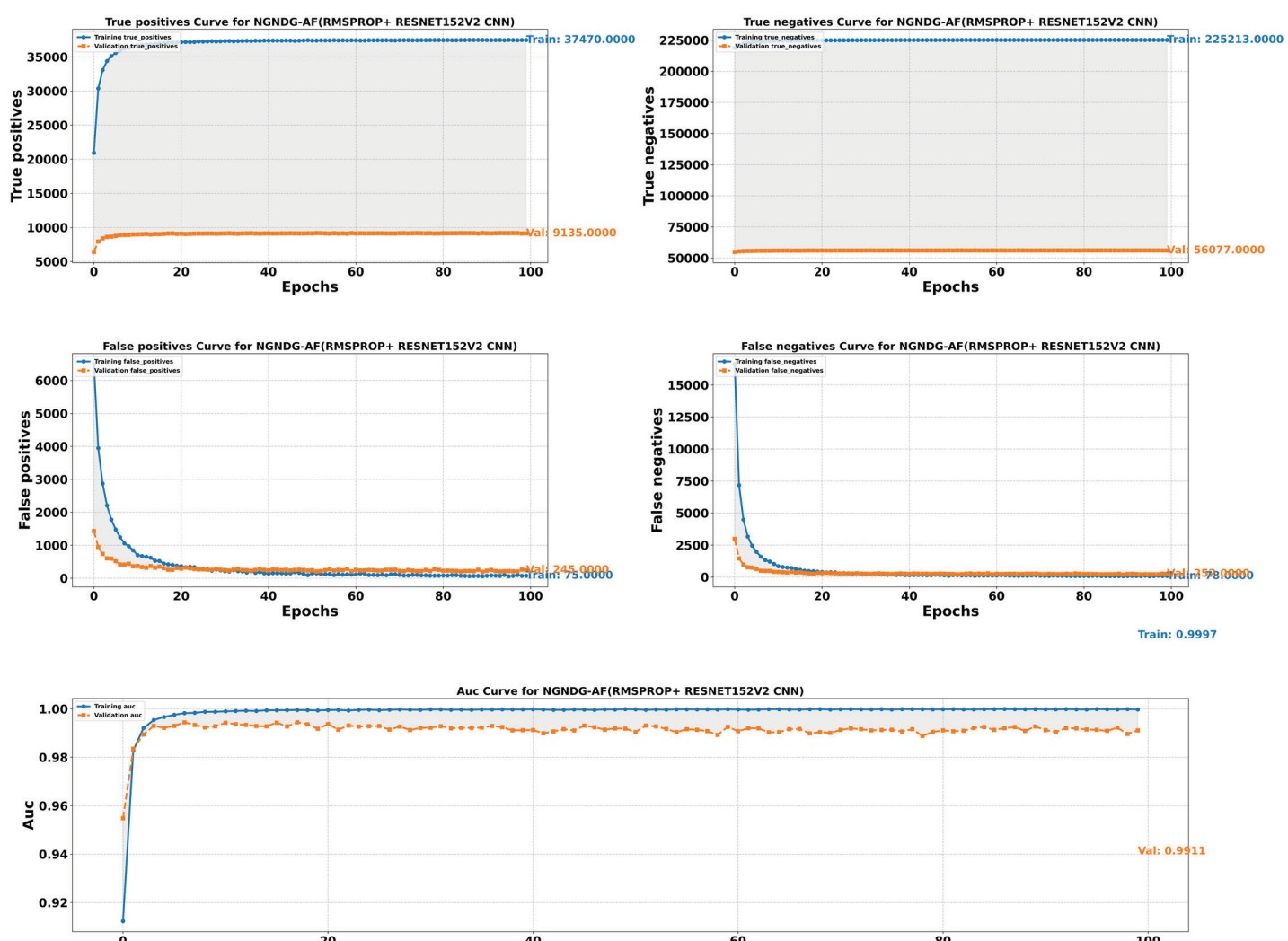

**Fig 9. False/True positive and negative of ResNet152V2 CNN with Ramsprop for $\beta = 1$.**

efficient learning in this instance as that which accompanies both high performance and a low computational load. In conclusion, an optimized CNN architecture using the aforementioned NGNDG-AF would result in high performance for resource-aware applications.

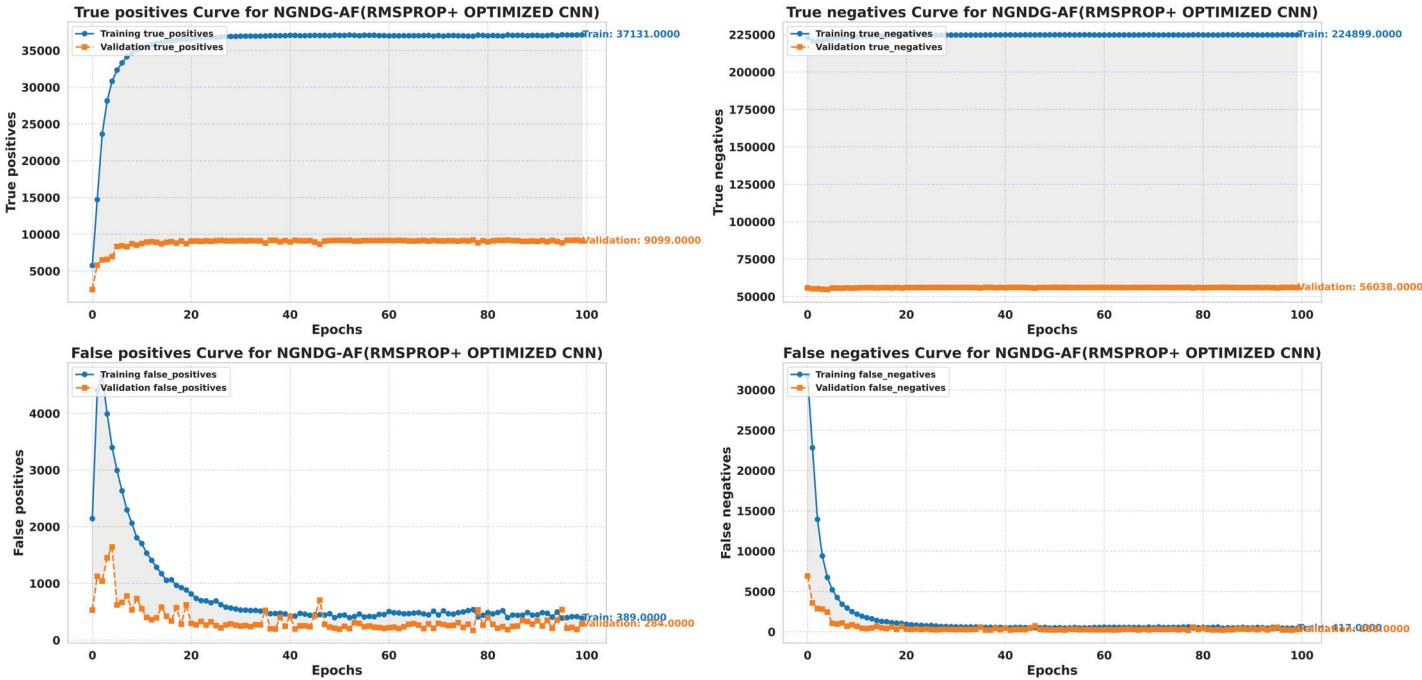

**Fig 10. False/True positive and negative of optimized CNN with Ramsprop $\beta = 1$.**

The Table 6 and Figs 9, 10, 11, 12 provided the overall performance metrics of true positives(TP), true Negatives(TN), false positives(FP), and false negatives(FN) for the optimized CNN and ResNet152V2 models underneath specific optimization algorithms (Adam and Rmsprop), all the usage of the NGNDG-AF activation function. These metrics are evaluated on each the education and validation units. Looking on the consequences, it is obtrusive that each the optimized CNN and ResNet152V2 models finished excessive numbers of true positives and true negatives, indicating their capacity to correctly classify instances from every positve and negative classes. Additionally, every model established low numbers of false positives and false negatives, suggesting that they make correct predictions whilst minimizing misclassifications. Under the Adam optimizer, the optimized CNN famous barely higher numbers of true Positives and true Negatives as compared to ResNet152V2. This indicated that the optimized CNN is barely greater powerful in effectively identifying each tremendous and poor times. Similarly, under the Rmsprop optimizer, the optimized CNN keeps a comparable overall performance to ResNet152V2 in phrases of true positives and true negatives, with a slightly better number of true positives.

The NGNDG-AF activation function possibly contributed considerably the NGNDG-AF activation feature in all likelihood contributed considerably to the performance of both models in efficaciously classifying instants. NGNDG-AF facilitated better gradient go together with the go along with the float within the direction of training, allowing the models to converge faster and keep away from problems like vanishing gradients. Additionally, NGNDG-AF helped seize complicated features within the information, fundamental to advanced kind accuracy. The advanced time-honored overall performance of the optimized CNN, specially in phrases of true positive and true negative counts, may be attributed to its precise form and the effective mixture of optimization set of rules and activation feature. The CNN's shape had some branches internal blocks that allowed better feature extraction, giving better ordinary

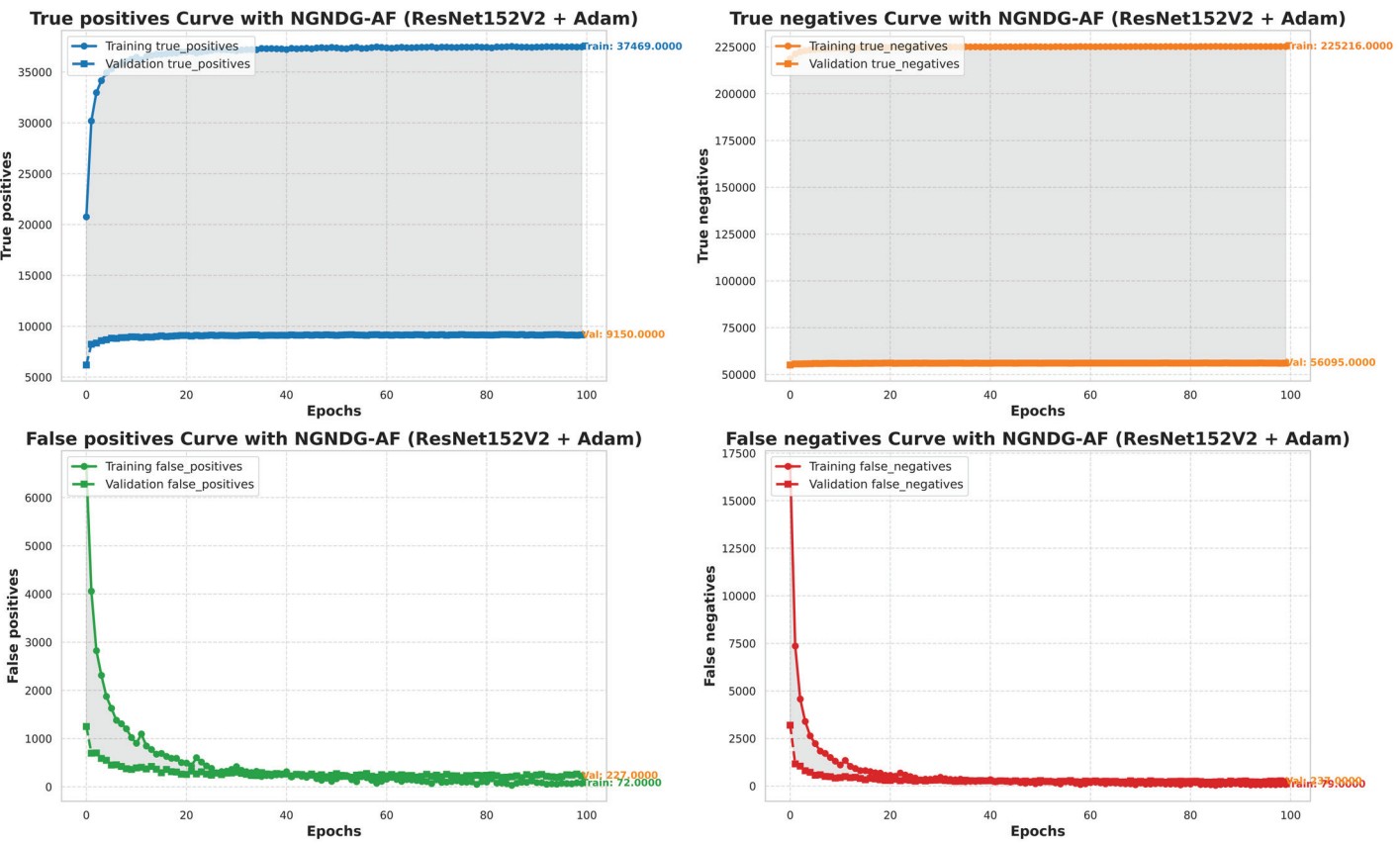

**Fig 11. False/True positive and negative of ResNet152V2 CNN with Adam for $\beta = 1$.**

average overall performance than ResNet152V2. Also, using NGNDG-AF activation superior the model's functionality to observe complex facts styles, developing accuracy and reducing misclassifications.

The Table 7 provided an evaluation on type techniques carried out to the HAM10000 dataset and highlighting their respective overall performance metrics such, as accuracy, precision, recall, F1 score and AUC. Each method has its strengths and weaknesses in categorizing skin lesion images. efficientnets reveal accuracy and high AUC indicating overall performance even as S-CNN suggests mild accuracy and F1 score. MobileNet and modified InceptionV4 exhibit first rate performance with variations in precision, recall and F1 score. Incremental CNN and decision support System show accuracy with change offs, in Recall and F1-score. Among the models that use pretrained architectures like ResNet50, Xception and a mixture of ResNet50, with VGG16 all showed high accuracy and spectacular AUC values demonstrating the effectiveness of these architectural designs. However it is ResNet152V2 that simply shines with overall performance across all assessment metrics showcasing its ability in correctly categorizing skin lesions. The optimized CNN model in addition raises the overall performance bar outperforming strategies in phrases of accuracy, precision, recall and F1-score with a slightly lower AUC compared to ResNet152V2. This first-rate overall performance may be attributed to the incorporation of the NGNDG AF activation characteristic, which enhanced gradient flow for the duration of training leading to steady and effective learning.

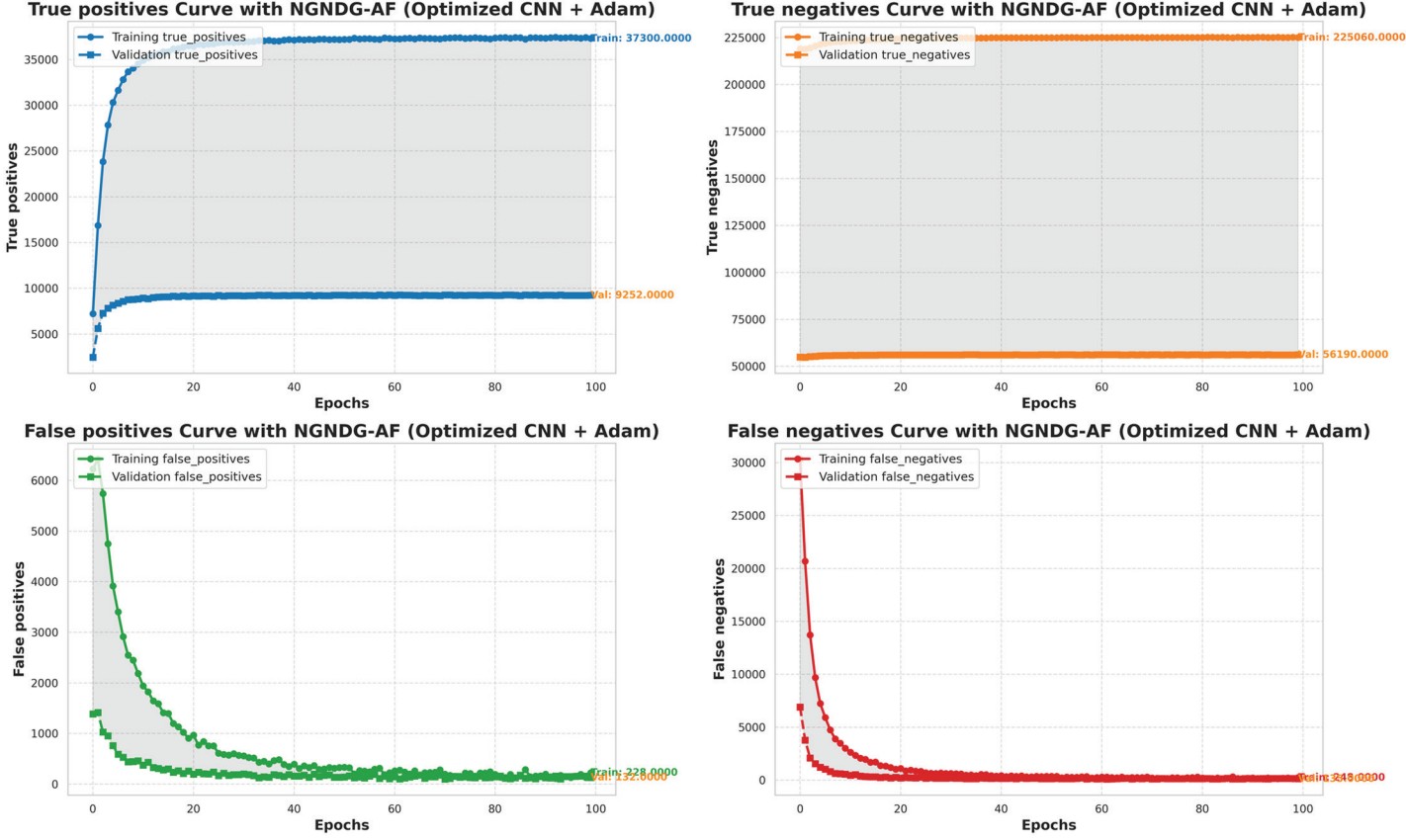

**Fig 12. False/True positive and negative of optimized CNN with Adam for $\beta = 1$.**

**Table 7. Quantitative results of different classification methods.**

| Model (HAM10000) | Accuracy | Precision | Recall | F1 Score | AUC |
|---|---|---|---|---|---|
| **ResNet50 and Xception** [46] | 0.9261 | 0.9207 | 0.9221 | 0.9261 | 0.9837 |
| **ResNet50 and VGG16** [46] | 0.9321 | 0.9292 | 0.9300 | 0.9321 | 0.9810 |
| **EfficientNets** [47] | ...... | 0.89 | 0.89 | ...... | 0.97 |
| **S-CNN** [48] | 0.80 | 0.86 | 0.82 | 0.85 | ...... |
| **MobileNet** [49] | 0.8315 | 0.89 | 0.83 | 0.83 | ...... |
| **Modified Inception-v4** [50] | 0.8617 | ...... | ...... | 0.717 | 0.88 |
| **Incremental CNN** [51] | 0.9026 | 0.89 | 0.89 | 0.88 | ...... |
| **Decision Support System** [52] | 0.905 | 0.88 | 0.77 | 0.74 | ...... |
| **ResNet152v2 (NGNDG-AF)** | 0.9788 | 0.9758 | 0.9748 | 0.9943 | 0.9941 |
| **Optimized CNN (NGNDG-AF)** | 0.9857 | 0.9859 | 0.9856 | 0.9996 | 0.9979 |

Comparison of various classification method using HAM10000 dataset: accuracy, precision, recall, F1-score and AUC.

The intrinsic properties of NGNDG AF played a position in maintaining information in networks enabling the optimized CNN model to achieved state of the artwork consequences in photo classification responsibilities such, as those related to the HAM10000 dataset.

The Figs 15, 16, 17, 18, 19, 20 are the evaluation of the optimized CNN system and the ResNet152V2 CNN system with Adam and Rmsprop revealed their respective advantages

**Table 8. Hyperparameters for CNN models**

| Parameters | ResNet152V2 | Optimized CNN |
|---|---|---|
| Image Size | $28 \times 28 \times 3$ | $28 \times 28 \times 3$ |
| Batch Size | 128 | 128 |
| Learning Rate | 0.001 | 0.001 |
| Decay Factor | 0.5 | 0.5 |
| Patience | 2 | 2 |
| Optimizer | Adam and Rmsprop | Adam and Rmsprop |
| Epochs | 100 | 100 |

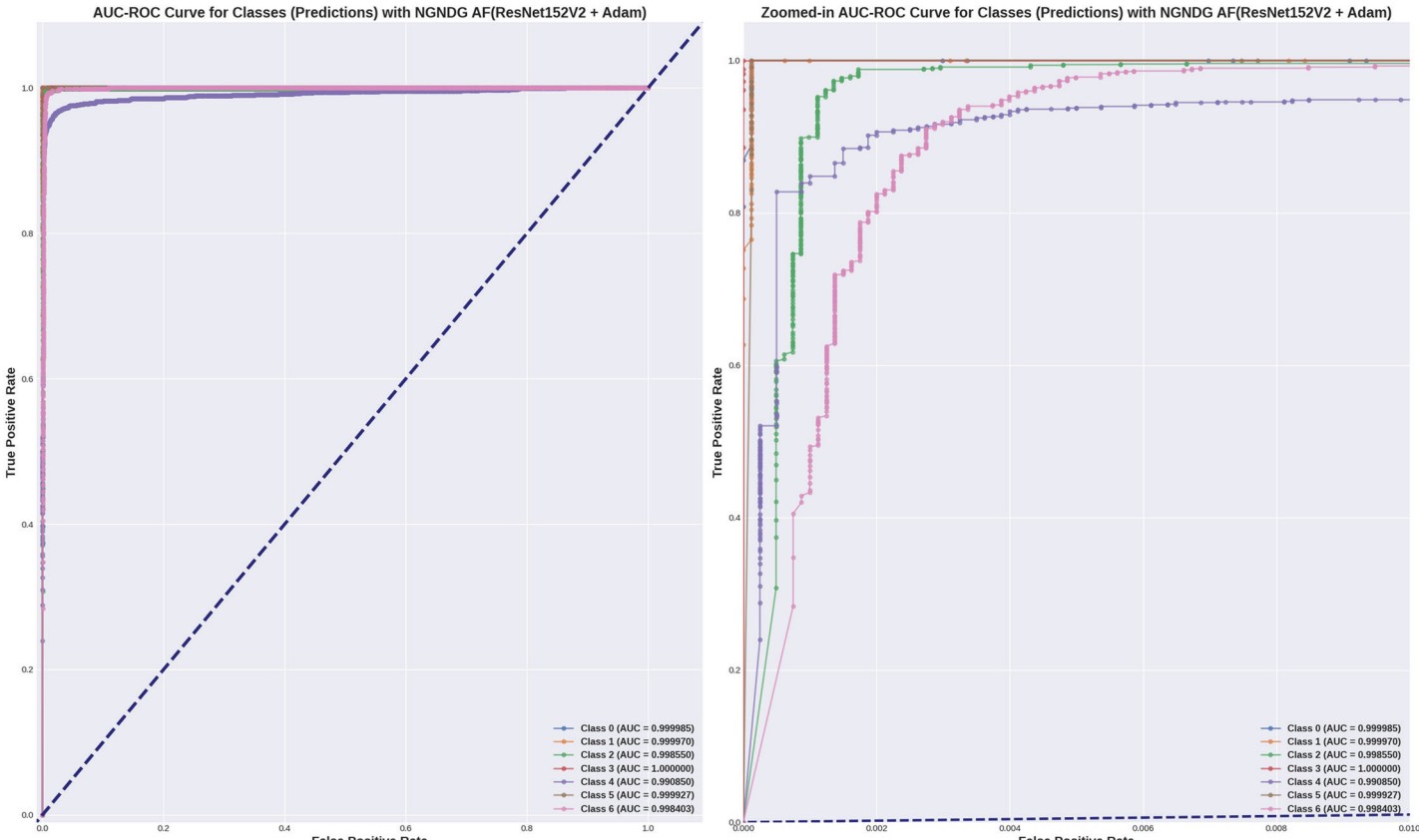

**Fig 13. AUC of ResNet152V2 CNN with Adam for $\beta = 1$.**

and disadvantages across a true one in each class of skin lesions on the seen and unseen test data of the HAM1000 dataset. Confusion matrix gave performance of classification systems for each class of HAM10000 skin cancer data set. There exist much more misclassifiction for class '4' as class '2' and '6' but for other class described classification systems in this work are showing extra ordinary prediction performance. Each class of given data is classified more correctly. However, the present study does not apply some mechanisms such as regularization and data augmentation that would be useful in measuring how well the model generalizes to unseen data. Holdout validation and dropout rates are two components of the robust valida- tion strategy that we employed. Which also suggests that the optimized CNN is generalizing well to new data that has not been considered before. When enhanced with Adam, the CNN

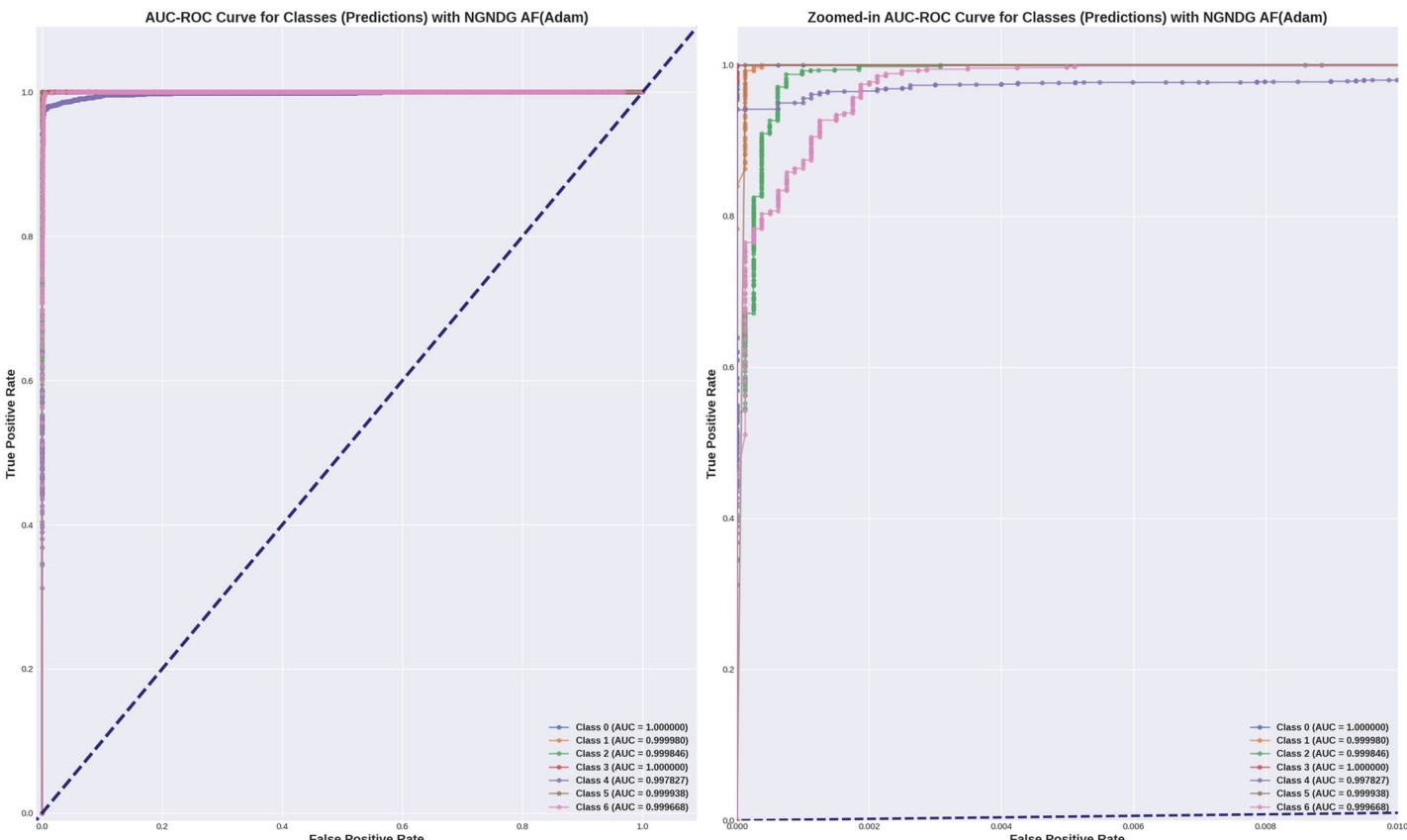

**Fig 14. AUC of optimized CNN with Adam for $\beta = 1$.**

system was further improved so that excellent predictions is be made. It is the best at guessing pictures of basal cell cancer, bumpy skin growths, melanoma, blood vessel problems, moles, and rough skin areas. The system achieved high accuracy overall in many tests. This shows it can pick out hard details in skin photos really well. In assessment, while ResNet152V2 CNN System with adam commonly performed properly, it exhibited some variability in its predictions within sure instructions. Although reaching excessive accuracy quotes for most instances, ResNet152V2 now and again struggles with unique pix inside lessons like basal cell carcinoma, dermatofibroma, melanoma, vascular lesion, and melanocytic nevi. This variability suggested capacity barriers within the model's capacity to generalize efficaciously throughout instances of the same class. It is feasible that the architectural layout of ResNet152V2, in spite of its intensity and complexity, won't fully seize the various manifestations of skin lesions gift in the dataset. The excellent results delivered by the optimized CNN system are due to a unique activation characteristic NGNDG-AF that was applied to it. This characteristic manages nonlinearities in data with considerable accuracy, which leads to more exact predictions. it was found that the Adam-optimized CNN system has outperformed the Adam-ResNet152V2 CNN System in terms of accuracy and precision when using skin lesion data from the HAM1000 dataset after multiple training sessions.

In Figs 21, 22, 23, 24 the precised integration of NGNDG-AF with Grad-CAM and Grad-CAM++ showcased the profound have an impact on of the feature's gradient on visualizing important areas internal pores and skin cancer medical photos. The specific properties

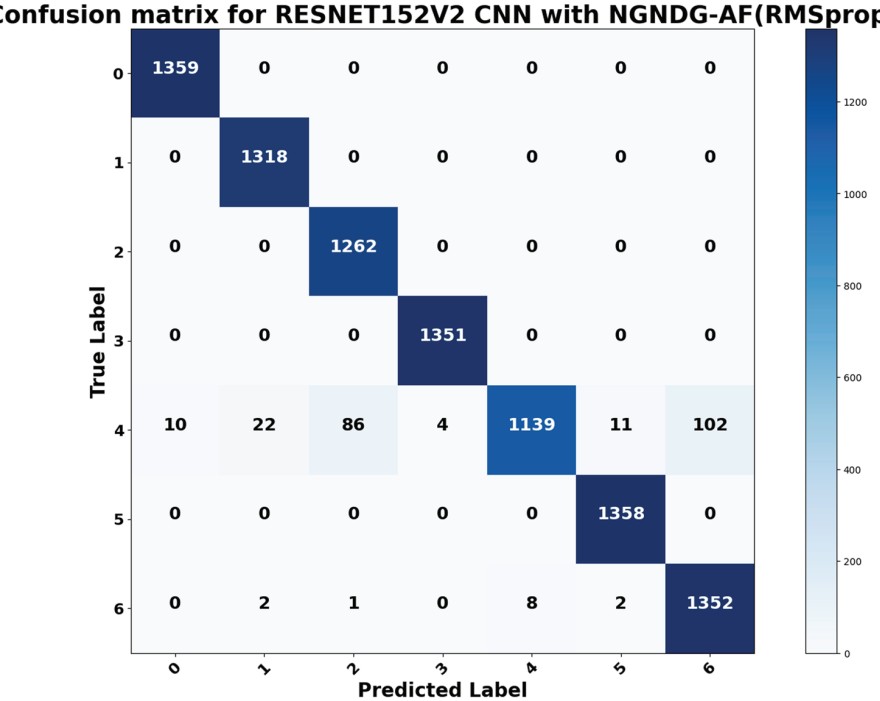

**Fig 15. Confusion matrix of ResNet152V2 CNN with Rmsprop for $\beta = 1$.**

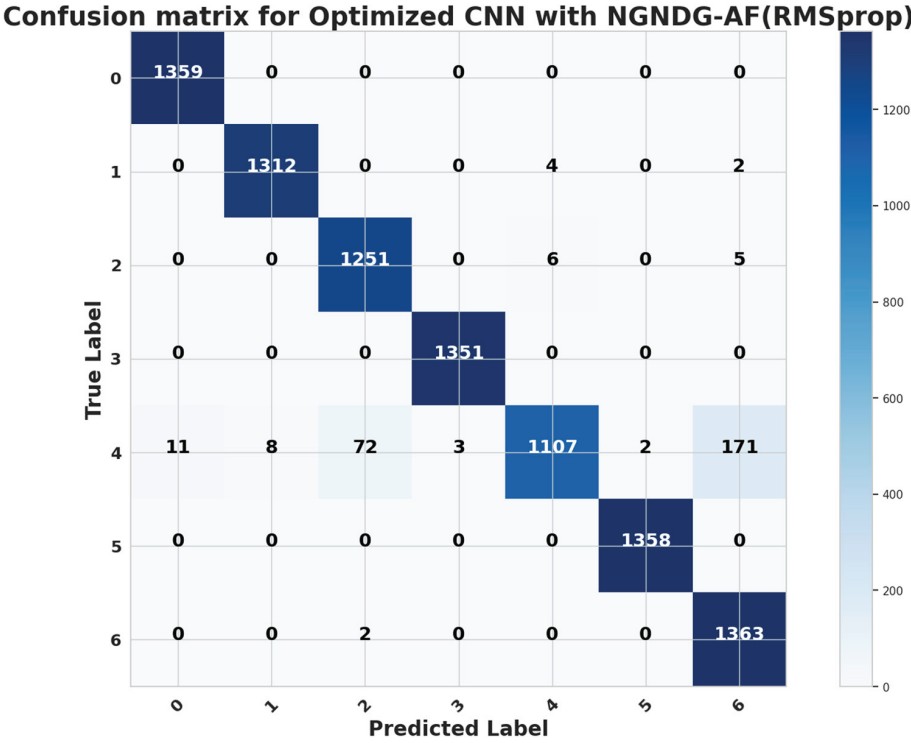

**Fig 16. Confusion matrix of optimized CNN with Rmsprop for $\beta = 1$.**

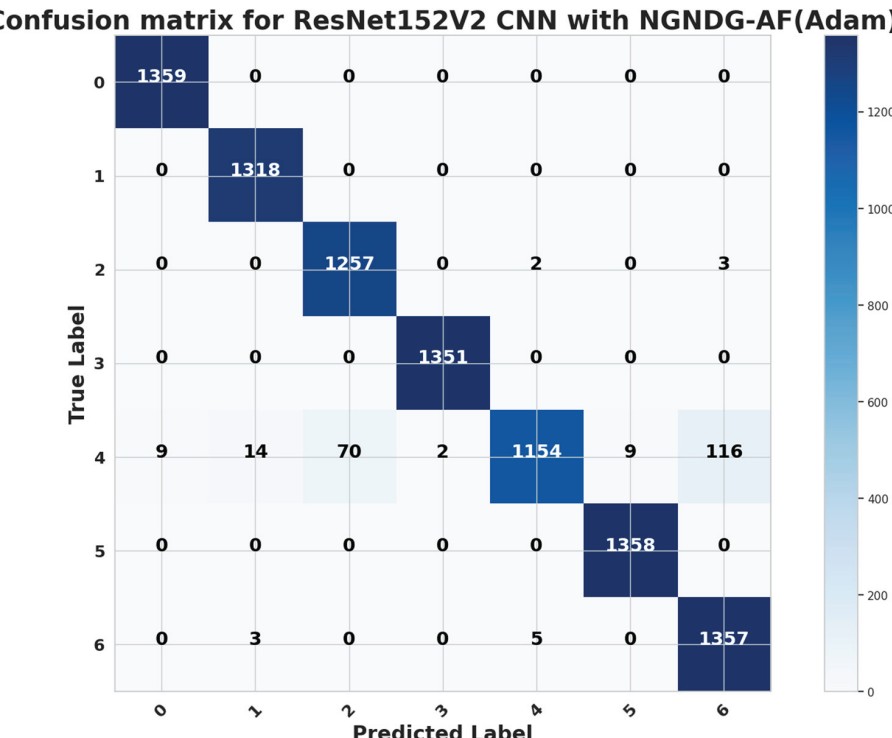

**Fig 17. Confusion matrix of ResNet152V2 CNN with Adam for $\beta = 1$.**

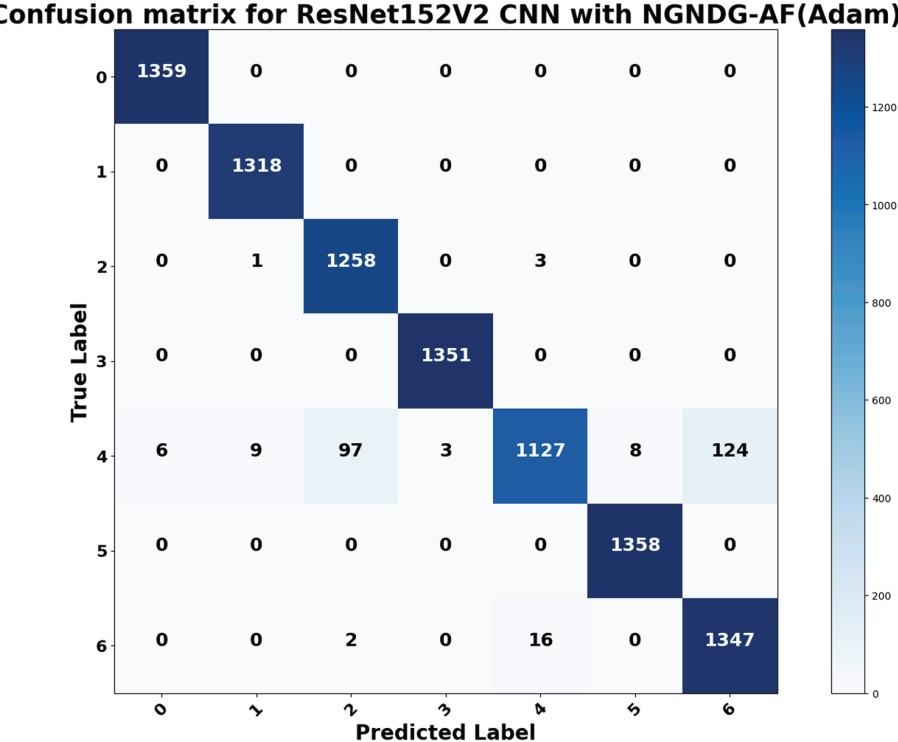

**Fig 18. Confusion matrix of optimized CNN with Adam for $\beta = 1$.**

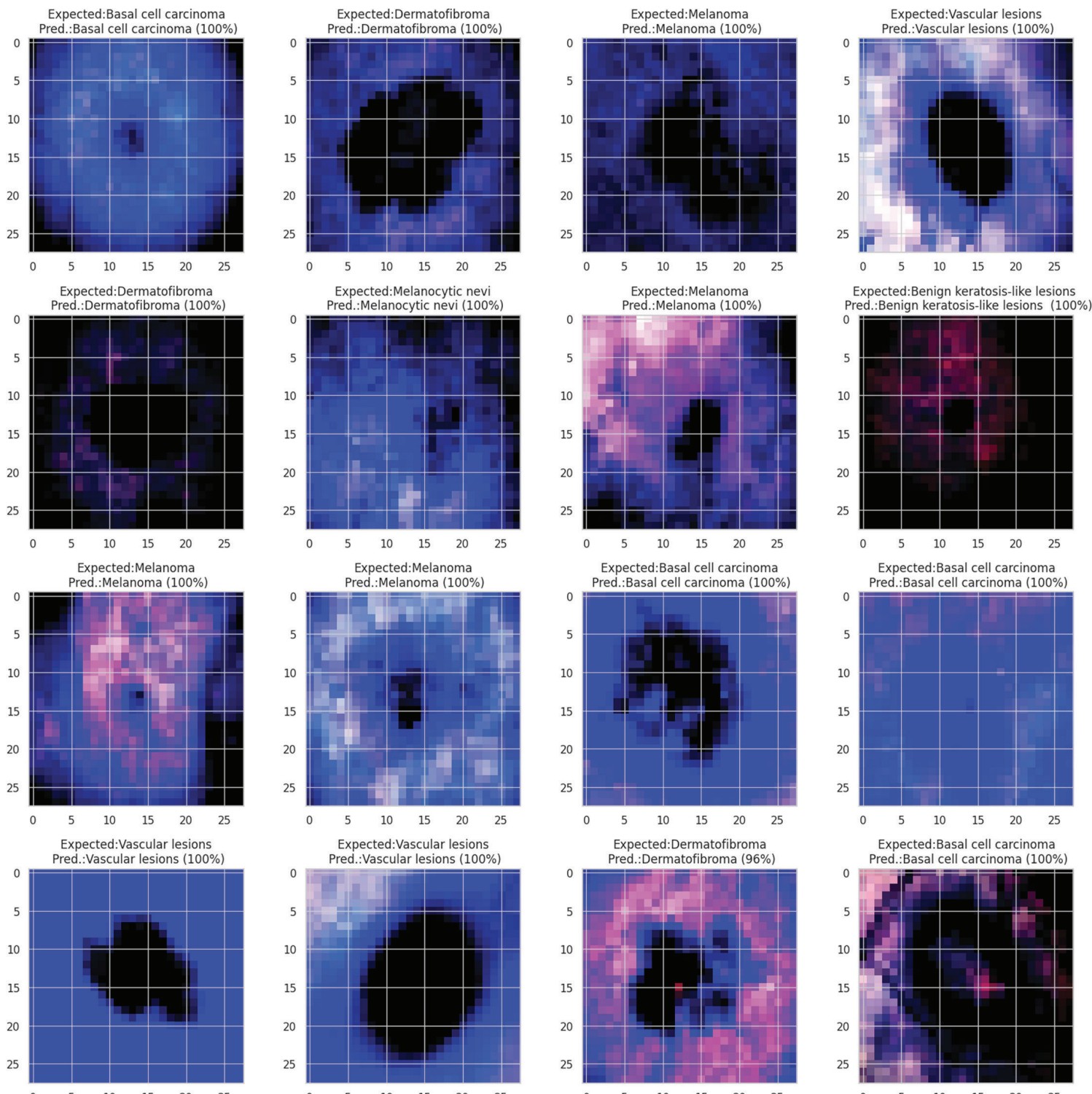

**Fig 19. Cross validation of skin cancer data with prediction accuracy of optimized CNN.**

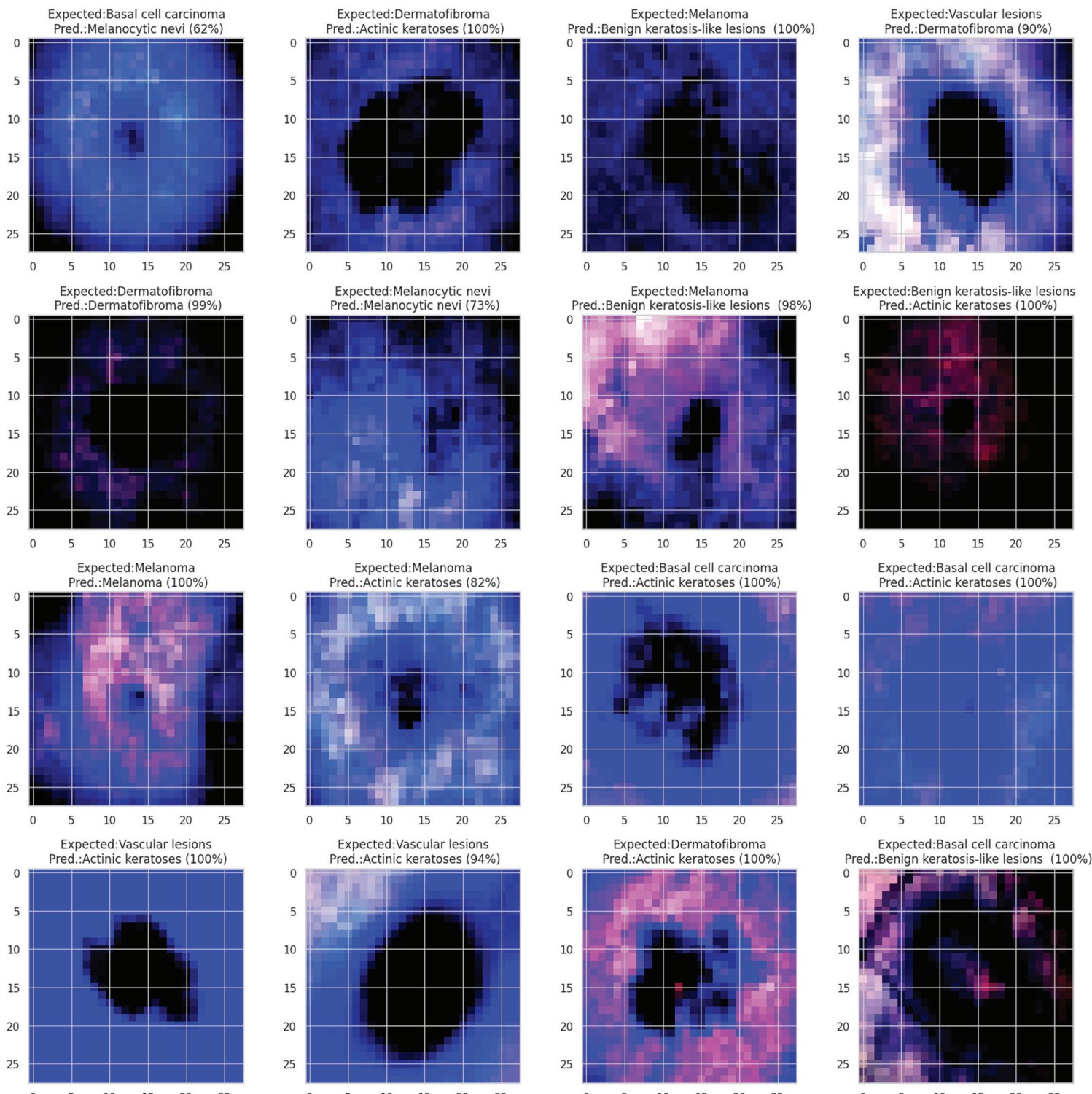

**Fig 20. Cross validation of skin cancer data with prediction accuracy of ResNet152V2 CNN.**

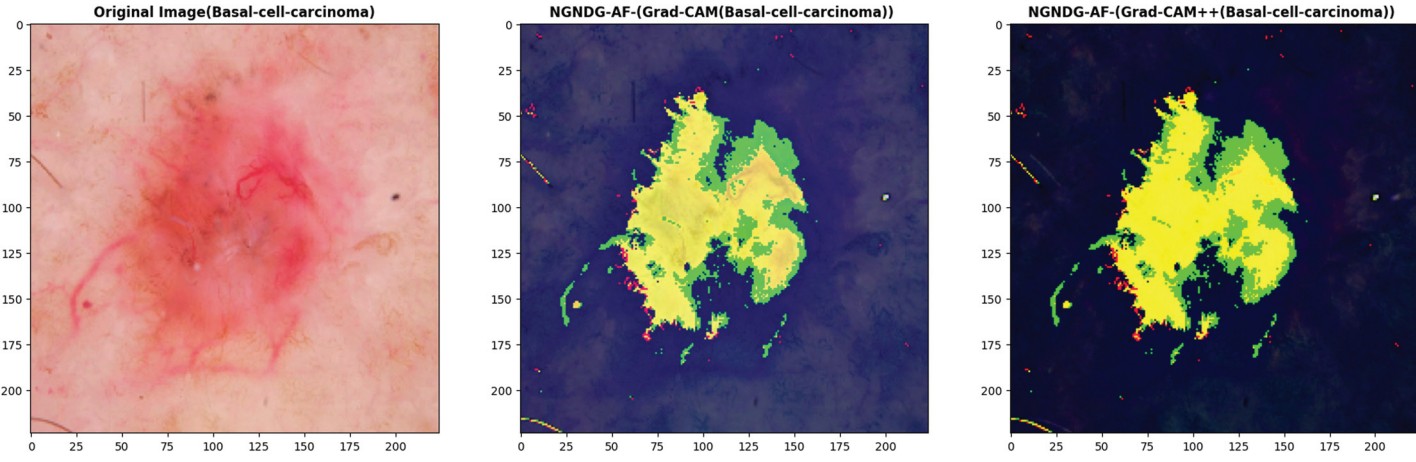

**Fig 21. Grad- CAM and Grad-Cam++ visualization of optimized CNN.**

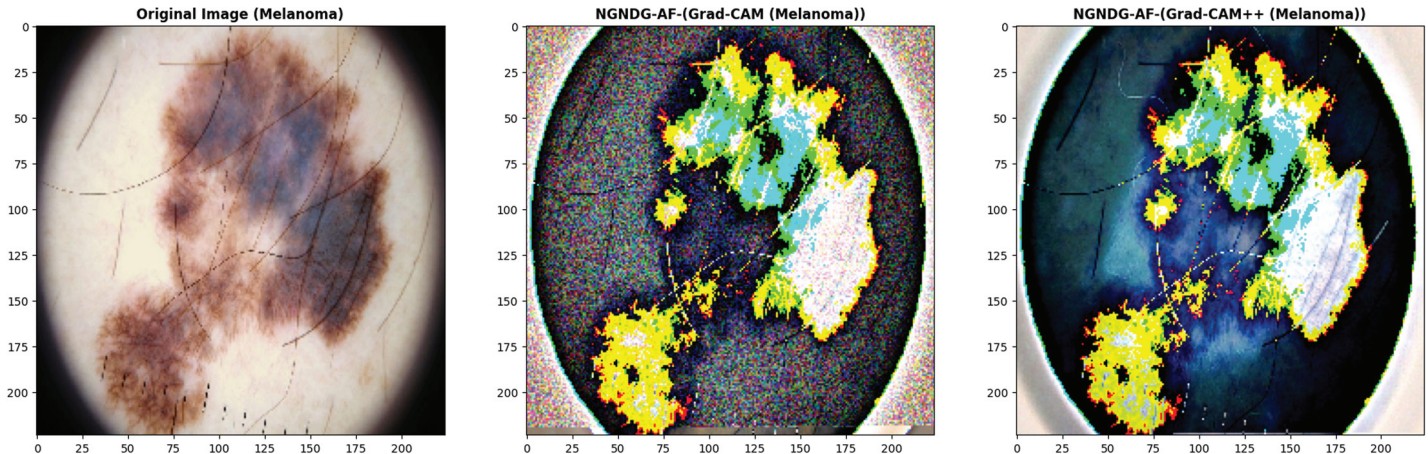

**Fig 22. Grad- CAM and Grad-Cam++ visualization of optimized CNN.**

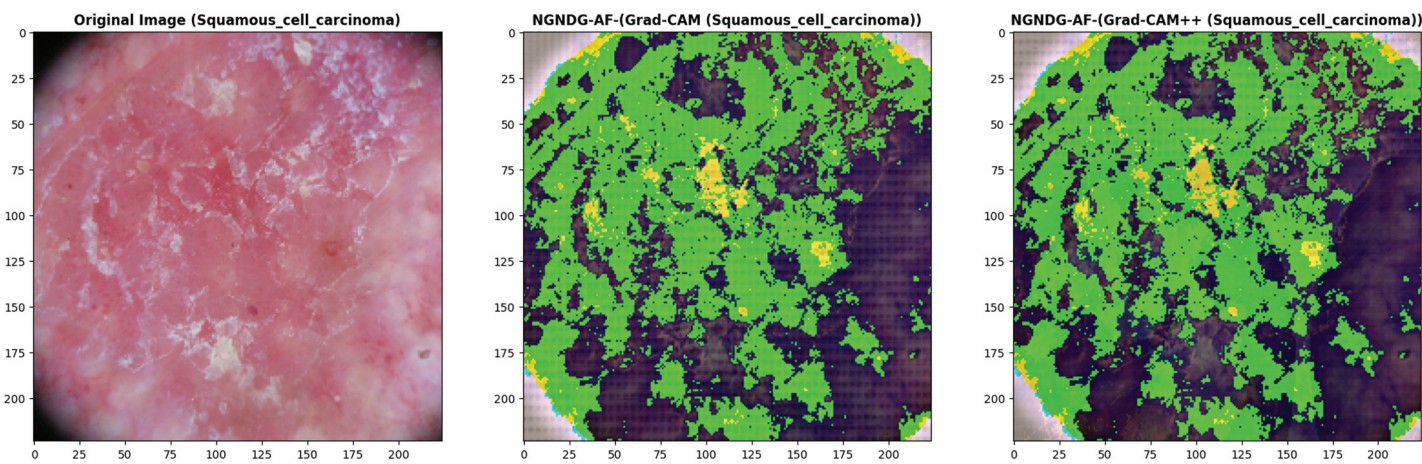

**Fig 23. Grad- CAM and Grad-Cam++ visualization of optimized CNN.**

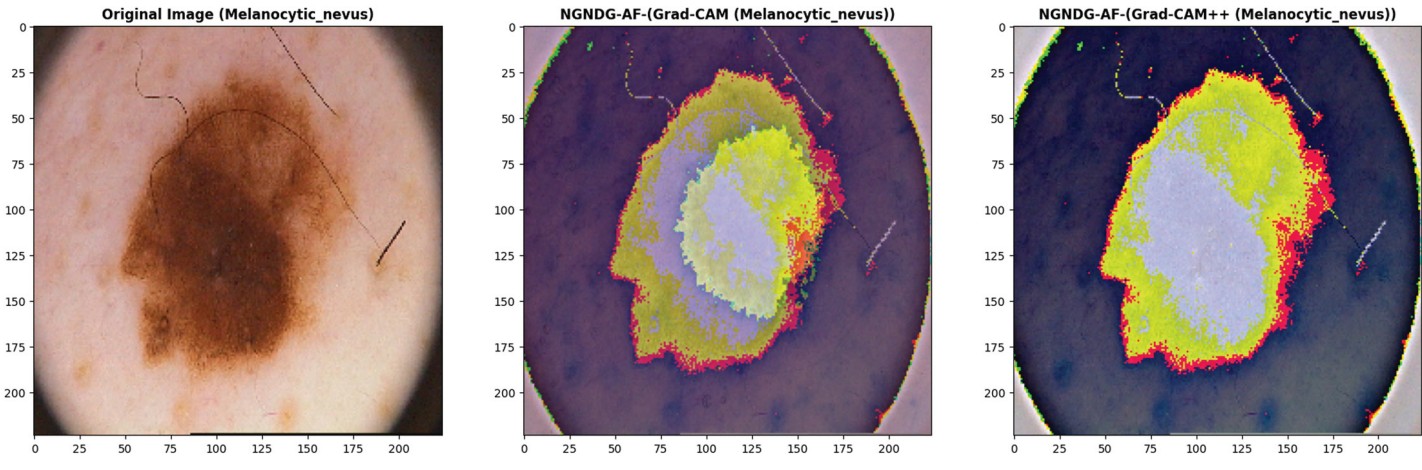

**Fig 24. Grad- CAM and Grad-Cam++ visualization of optimized CNN.**

of NGNDG-AF, mainly its derivative, play a pivotal feature in improving the general performance of each Grad-CAM and Grad-CAM++. The gradient of NGNDG-AF serves as a crucial thing in figuring out the significance and relevance of features extracted with the aid of the CNN. By leveraging the derivative of NGNDG-AF, Grad-CAM and Grad-CAM++ efficiently picked out and emphasized the most influential areas within the enter images, thereby producing heatmaps which is extra correct, localized, and aligned with the model's decision-making manner. The better-order derivatives of NGNDG-AF enabled Grad-CAM and Grad-CAM++ to seized extra difficult patterns and diffused nuances inside the data, resulting in a greater subtle and granular visualization of "searing" areas contributing to the CNN's choices. The derivative of NGNDG-AF holds precise significance in clinical records wherein regions showed off sharp modifications in each shade and structure. This derivative is notably touchy to Grad-CAM and Grad-CAM++ for those searing areas, making sure that these strategies appropriately become aware of and emphasize the important regions wherein the CNN version's choice-making manner is mainly inspired by using abrupt transitions or great variations in the enter facts. The significance of the derivative lies in its potential to provide gradient-based localization, improved function discrimination, and stepped forward sensitivity to modifications in the enter information, all of which are essential for the accurate visualization of essential areas by means of Grad-CAM and Grad-CAM++. The synergistic mixture of NGNDG-AF, with its advanced gradient homes, and the visualization competencies of Grad-CAM and Grad-CAM++ culminates in a visible illustration this is both interpretable and informative. This novel integration not only improves the model's accuracy and reliability, however it also develops greater consider and information amongst stakeholders, inclusive of scientists and sufferers. Immediately, provided a entire seen example of ways the gradient of NGNDG-AF improved the general overall performance of Grad-CAM and Grad-CAM++ in visualizing essential places inside scientific photos. This present day integration emphasized the significance of making use of more advantageous activation function and visualization strategies to progress the sphere of clinical photo evaluation in medical field.

## 10 Conclusions

This research introduced a optimized approach to the improvement of skin cancer screening in smart healthcare environments by blending CNN with NGNDG-AF and XAI. Skin Cancer

Diagnosis is an important public health challenge worldwide that requires accurate and efficient solutions. Therefore, traditional diagnostic techniques often experience delays and subjectivity, necessitating innovative approaches. Our investigation responded to this urgency by developing NGNDG-AF activation function which improved the activation process in CNNs making them more selective. With application of the optimized CNN architecture like optimized CNN built upon HAM10000 dataset, high classification accuracy rate has been attained on both training set (training accuracy 99%) and validation set (validation accuracy 98%). In addition, NGNDG-AF improved gradient flow during training phases and captured minute details from skin cancer images thus resulting in good performance. Additionally, XAI techniques involving Grad-CAM equipped with NGNDG-AF and Grad-CAM++ assisted in understanding model decisions leading to enhanced interpretability as well as trust among medical practitioners. In this way, medical images could be used to identify the essential parts of the human body that are significant for classifying patients with different diseases or conditions. Comparing this optimized CNN model with other alternative optimization algorithms and architectures clearly showed how much better it performed than any other method of optimizing CNN. In addition, the optimized CNN had better performance metrics such as precision (98.59%), recall (98.56%), and AUC (99.79%). Studying classifying CNN system with high value of precision is classifying images correctly labeled with few false positive. This high value of recall ensures that most suitable images are picked. while a high AUC demonstrates the model's strong ability to distinguish between different image classes across thresholds. The evaluation of the optimized CNN system with Adam and the ResNet152V2 CNN system with Adam revealed their respective advantages and disadvantages across a true one in each class of skin lesions on the unseen test data of the HAM1000 dataset. Hold-out validation and dropout rates are two components of the robust validation strategy that we are employing. This method has enabled the achievement of high validation accuracy, which also suggests that the model generalizes well to new data that has not been considered before. This implies that it is more effective in classifying skin-lesions accurately. In addition, this method provides results that are faster, more accurate, and easier for non-dermatologists to use. In addition, NGNDG-AF integration with XAI techniques improves smart healthcare applications while fostering trust and comprehension about this ai-driven medical diagnosis. The rapidity and certainty of our model's diagnoses would have profound consequences for detecting and treating disease early, the next research efforts will seek to explore the attention mechanism and transformers integration in further improving interpretability and performance.

## Author contributions

**Conceptualization:** Ali Raza, Sami Ullah.

**Data curation:** Akhtar Ali.

**Formal analysis:** Sami Ullah.

**Investigation:** Yasir Nadeem Anjum.

**Validation:** Basit Rehman.

**Writing – original draft:** Akhtar Ali.

**Writing – review & editing:** Akhtar Ali.

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
