## [Decision Letter · Decision Letter 0]

8 Oct 2024

PONE-D-24-40790Optimizing Skin Cancer Screening with Convolutional Neural Networks in Smart Healthcare SystemsPLOS ONE

Dear Dr. Ali,

Thank you for submitting your manuscript to PLOS ONE. After careful consideration, we feel that it has merit but does not fully meet PLOS ONE’s publication criteria as it currently stands. Therefore, we invite you to submit a revised version of the manuscript that addresses the points raised during the review process.

We look forward to receiving your revised manuscript.

Kind regards,

Asadullah Shaikh, Ph.D.

Academic Editor

PLOS ONE

Journal Requirements:

1. When submitting your revision, we need you to address these additional requirements. Please ensure that your manuscript meets PLOS ONE's style requirements, including those for file naming. The PLOS ONE style templates can be found at https://journals.plos.org/plosone/s/file?id=wjVg/PLOSOne_formatting_sample_main_body.pdf and https://journals.plos.org/plosone/s/file?id=ba62/PLOSOne_formatting_sample_title_authors_affiliations.pdf 2. Please note that PLOS ONE has specific guidelines on code sharing for submissions in which author-generated code underpins the findings in the manuscript. In these cases, we expect all author-generated code to be made available without restrictions upon publication of the work. Please review our guidelines at https://journals.plos.org/plosone/s/materials-and-software-sharing#loc-sharing-code and ensure that your code is shared in a way that follows best practice and facilitates reproducibility and reuse. 3. Please provide a complete Data Availability Statement in the submission form, ensuring you include all necessary access information or a reason for why you are unable to make your data freely accessible. If your research concerns only data provided within your submission, please write "All data are in the manuscript and/or supporting information files" as your Data Availability Statement. 4. PLOS requires an ORCID iD for the corresponding author in Editorial Manager on papers submitted after December 6th, 2016. Please ensure that you have an ORCID iD and that it is validated in Editorial Manager. To do this, go to ‘Update my Information’ (in the upper left-hand corner of the main menu), and click on the Fetch/Validate link next to the ORCID field. This will take you to the ORCID site and allow you to create a new iD or authenticate a pre-existing iD in Editorial Manager.

Reviewers' comments:

Reviewer's Responses to Questions

**Comments to the Author**

1. Is the manuscript technically sound, and do the data support the conclusions?

Reviewer #1: Partly

Reviewer #2: Partly

Reviewer #3: Yes

2. Has the statistical analysis been performed appropriately and rigorously? 

Reviewer #1: Yes

Reviewer #2: Yes

Reviewer #3: Yes

3. Have the authors made all data underlying the findings in their manuscript fully available?

Reviewer #1: Yes

Reviewer #2: No

Reviewer #3: Yes

4. Is the manuscript presented in an intelligible fashion and written in standard English?

Reviewer #1: Yes

Reviewer #2: Yes

Reviewer #3: Yes

5. Review Comments to the Author

Reviewer #1: 1. The abstract of the proposed model needs to concise and brief in order to clearly demonstrate the whole manuscript.

2. The related papers in the literature review needs to critically analyzed.

3. The quality of the figures are poor, the authors needs to revised in 600 dpi for its clear visualizations.

4. How the authors handle the generalization of the proposed model.

5. For the reader concerns and to provide more details related to deep learning models, I suggest mentioning the recent deep predictors such as AIPs-SnTCN, StackedEnC-AOP, DeepAVP-TPPred, iAFPs-Mv-BiTCN, and Deepstacked-AVPs.

6. It seems like that the authors pasted the images of the table instead creating the new table.

7. The conclusion needs more improvement, the word “the proposed” used so many times.

Reviewer #2: Abstract Section: Abstract should be robust in accordance to the brief introduction of the domain and topic which leads to the proposed methodology. Lastly, Research results should also be discussed briefly in comparison of state of the art.

Language and Grammar: There are some language and grammatical errors, which should be corrected to maintain the manuscript's academic integrity.

Technical Terminology: The use of technical jargon is at times inconsistent. Ensuring consistent use of technical terms would improve the manuscript's readability.

Graphics and Visual Aids: Figures and diagrams in the manuscript could be enhanced for clarity and better understanding. Furthermore there should be proper explanation against each figure.

Comparative Evaluations: Authors should include the section of comparative evaluations and they need to validate their finding with the literature.

Formatting and Structure: There are many inconsistencies in the formatting and structure of the manuscript that could be rationalized for a more professional presentation. Especially, the headings and numbering should be streamlined.

Explanation of Figures: Additional explanations or captions for all figures would enhance their relevance and clarity in the context of the text.

Reviewer #3: The manuscript presents a valuable contribution to the field of medical imaging, particularly in the area of skin cancer detection using deep learning models. The introduction of the NGNDG-AF activation function and the integration of explainable AI techniques, such as Grad-CAM, provide significant advancements in both accuracy and transparency for the proposed model. The results are promising, showcasing high accuracy in skin cancer classification with the potential to positively impact clinical decision-making. However, there are several aspects that could be strengthened to further improve the robustness, clarity, and generalizability of the work.

Several key areas need more attention to strengthen the claims made in the manuscript:

1.While NGNDG-AF is introduced as a new activation function, a more detailed mathematical analysis is required to justify its advantages over existing functions like ReLU and ELU. Expanding the experimental comparisons with a broader range of activation functions could better demonstrate its superiority.

2. The study is primarily focused on the HAM10000 dataset, which, while widely used, may not fully represent real-world variability in skin cancer. Including additional datasets, such as ISIC 2019, would enhance the generalizability of the results and provide stronger evidence of the model's robustness.

3. While the results are comprehensive, they could benefit from clearer organization. The improve visibility of confusion matrices would visually enhance the presentation of model performance.

4. Although the validation accuracy is high, the paper lacks a detailed discussion of overfitting risks. More attention to techniques like dropout rates, regularization, or data augmentation would improve the reliability of the results and demonstrate the model's ability to generalize well to unseen data.

5. The development of a smart healthcare system via an Android application is a noteworthy goal, but this aspect is only briefly mentioned. More details regarding the app’s architecture, usability, and implementation would provide a stronger case for its practical applicability in a healthcare setting.

6. PLOS authors have the option to publish the peer review history of their article (what does this mean?). If published, this will include your full peer review and any attached files.

Reviewer #1: No

Reviewer #2: **Yes: **Awais Ahmad

Reviewer #3: **Yes: **Muhammad Ahmad Pasha

---

## [Author Response · Author response to Decision Letter 1]

21 Oct 2024

We have uploaded the thoroughly revised version according to the above-mentioned comments, suggestions and recommendations.

---

## [Decision Letter · Decision Letter 1]

19 Nov 2024

PONE-D-24-40790R1Optimizing Skin Cancer Screening with Convolutional Neural Networks in Smart Healthcare SystemsPLOS ONE

Dear Dr. Ali,

Thank you for submitting your manuscript to PLOS ONE. After careful consideration, we feel that it has merit but does not fully meet PLOS ONE’s publication criteria as it currently stands. Therefore, we invite you to submit a revised version of the manuscript that addresses the points raised during the review process.

We look forward to receiving your revised manuscript.

Kind regards,

Asadullah Shaikh, Ph.D.

Academic Editor

PLOS ONE

Journal Requirements:

Reviewers' comments:

Reviewer's Responses to Questions

**Comments to the Author**

1. If the authors have adequately addressed your comments raised in a previous round of review and you feel that this manuscript is now acceptable for publication, you may indicate that here to bypass the “Comments to the Author” section, enter your conflict of interest statement in the “Confidential to Editor” section, and submit your "Accept" recommendation.

Reviewer #1: All comments have been addressed

Reviewer #2: (No Response)

Reviewer #3: (No Response)

2. Is the manuscript technically sound, and do the data support the conclusions?

Reviewer #1: Yes

Reviewer #2: Yes

Reviewer #3: (No Response)

3. Has the statistical analysis been performed appropriately and rigorously? 

Reviewer #1: Yes

Reviewer #2: Yes

Reviewer #3: (No Response)

4. Have the authors made all data underlying the findings in their manuscript fully available?

Reviewer #1: Yes

Reviewer #2: Yes

Reviewer #3: (No Response)

5. Is the manuscript presented in an intelligible fashion and written in standard English?

Reviewer #1: Yes

Reviewer #2: Yes

Reviewer #3: (No Response)

6. Review Comments to the Author

Reviewer #1: My previous comments are successfully incorporated and now the paper has been significantly improved I think the paper can accepted from my side.

Reviewer #2: Figures and tables are not still propoerly formatted. There is severe visibility problems

Reviewer #3: (No Response)

7. PLOS authors have the option to publish the peer review history of their article (what does this mean?). If published, this will include your full peer review and any attached files.

Reviewer #1: No

Reviewer #2: **Yes: **Awais Ahmad

Reviewer #3: **Yes: **Muhammad Ahmad Pasha

---

## [Author Response · Author response to Decision Letter 2]

28 Nov 2024

Reviewer #2:

Reviewer Comment: Figures and tables are not still properly formatted. There is severe visibility problems

Author Response: The figures have been formatted according to the PLOS ONE standards using the platform provided at https://pacev2.apexcovantage.com/, and the tables have been formatted according to the standard table format specified by PLOS ONE.

Journal Requirements:

Author Response: The following changes in reference list have been made for the manuscript:

• References 1 , 16, 22, 27

---

## [Decision Letter · Decision Letter 2]

23 Dec 2024

Optimizing Skin Cancer Screening with Convolutional Neural Networks in Smart Healthcare Systems

PONE-D-24-40790R2

Dear Dr. Ali,

We’re pleased to inform you that your manuscript has been judged scientifically suitable for publication and will be formally accepted for publication once it meets all outstanding technical requirements.

Kind regards,

Asadullah Shaikh, Ph.D.

Academic Editor

PLOS ONE

Additional Editor Comments (optional):

Reviewers' comments:

Reviewer's Responses to Questions

**Comments to the Author**

1. If the authors have adequately addressed your comments raised in a previous round of review and you feel that this manuscript is now acceptable for publication, you may indicate that here to bypass the “Comments to the Author” section, enter your conflict of interest statement in the “Confidential to Editor” section, and submit your "Accept" recommendation.

Reviewer #4: All comments have been addressed

2. Is the manuscript technically sound, and do the data support the conclusions?

Reviewer #4: Yes

3. Has the statistical analysis been performed appropriately and rigorously? 

Reviewer #4: Yes

4. Have the authors made all data underlying the findings in their manuscript fully available?

Reviewer #4: Yes

5. Is the manuscript presented in an intelligible fashion and written in standard English?

Reviewer #4: Yes

6. Review Comments to the Author

Reviewer #4: (No Response)

7. PLOS authors have the option to publish the peer review history of their article (what does this mean?). If published, this will include your full peer review and any attached files.

Reviewer #4: No

---

## [Editor Report · Acceptance letter]

PONE-D-24-40790R2

PLOS ONE

Dear Dr. Ali,

I'm pleased to inform you that your manuscript has been deemed suitable for publication in PLOS ONE. Congratulations! Your manuscript is now being handed over to our production team.

Kind regards,

on behalf of

Prof. Asadullah Shaikh

Academic Editor

PLOS ONE